# FROM TWO TO ONE: HARMONIZING ATTENTION AND FEATURE DEBIASING FOR MULTIVARIATE TIME SERIES FORECASTING

## ABSTRACT

Multivariate time series forecasting (MTSF) models based on Transformers have demonstrated remarkable success in various applications, including energy management, weather forecasting, and traffic monitoring. However, due to the complex and intertwined correlations among variates, Transformer-based methods often fail to precisely model the interactions among series, leading to limited performance improvement. In this paper, we rigorously investigate and establish the phenomenon of feature oversmoothing in Transformer-based forecasters through a theoretical analysis. To this end, we then propose **FADformer**, a frequency-aware debiasing framework, which harmonizes the low- and high-frequency components of attention and feature maps to capture fine-grained patterns for accurate forecasting. Specifically, we design two plug-and-play modules using the Fourier transformation, where i) AttnDeb rescales high-frequency weights within attention modules to mitigate the low-pass limitation and ii) FeatDeb injects inductive feature bias into residual connections to amplify the important high-frequency signals. Extensive experiments on challenging real-world datasets show the superiority of our FADformer over existing state-of-the-art methods, in terms of both forecasting performance and generalization ability.

## 1 INTRODUCTION

Transformers have made significant breakthroughs in natural language processing (NLP) (Patel et al., 2023; Kedia et al., 2024) and computer vision (CV) (Crowson et al., 2024; Esser et al., 2024), and recently found their way into time series tasks (Eldele et al., 2024) as well. As one critical and widely studied task, multivariate time series forecasting (MTSF) has been extensively applied in the economic (Siami-Namini & Namin, 2018), meteorological (Wu et al., 2023b), and transportation (Yin et al., 2022) fields. The core challenge in building precise forecasting models is to effectively identify and mathematically characterize the intricate patterns within multivariate time series. By tokenizing time series with time points (Zhou et al., 2021), sub-series (Nie et al., 2023), or independent variates (Liu et al., 2024) and modeling their dependencies with the self-attention mechanism (Vaswani et al., 2017), various Transformer-based forecasters have shown impressive performance.

Despite their success, the effectiveness of capturing informative temporal variations and multivariate correlations remains a concern. On the one hand, real-world time series often exhibit heterogeneous temporal patterns caused by complex signals or waves (Lai et al., 2018; Li et al., 2019) that vary over time, such as low-frequency long-term periodicity or high-frequency fluctuation, regarded as important indicators for multivariate time series forecasting. On the other hand, researchers have recently identified a learning bias issue (Wang et al., 2022; Piao et al., 2024) in the Transformer, stemming from the low-pass filtering nature of the self-attention mechanism. This issue may also occur in time series forecasting, where the model unintentionally smooths out important and detailed patterns in feature space, resulting in unsatisfactory model performance.

To intuitively understand the complexity of time series data and the filtering nature of Transformer-based models, we present an ETTh2 case, which is accompanied by the corresponding visualization results in Figure 1. The upper figure of Figure 1(a) shows that different channels of ETTh2 (i.e., C1-C7) exhibit different temporal patterns across successive periods with inconsistent fluctua-

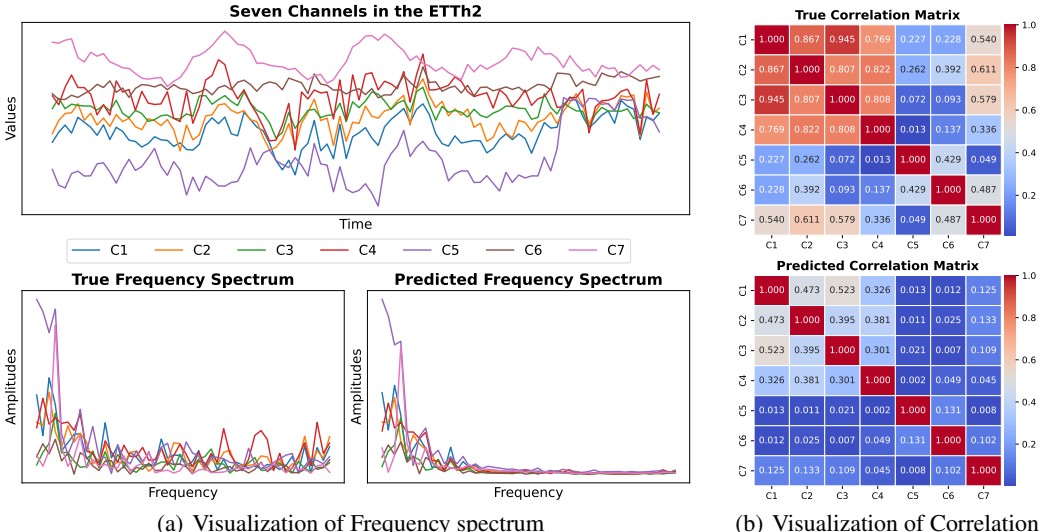

(a) Visualization of Frequency spectrum      (b) Visualization of Correlation

Figure 1: Visualization results of different channels (i.e., C1-C7) on ETTh2. The true spectrum and predicted spectrum are derived by transforming the ground truth and predicted results, respectively, in the frequency domain. The true correlation matrix is obtained by computing the correlations among the ground truth, while the predicted correlation matrix is calculated using the forecasting values obtained by iTransformer (Liu et al., 2024).

tions. The lower two subfigures display the frequency spectrum characteristics of the ground truths and forecasting results obtained using the self-attention mechanism. Comparatively, the true spectrum contains rich high-frequency information, whereas the predicted spectrum is smooth, lacking distinct fluctuations in high-frequency bands. Moreover, the visualizations of the true correlation matrix and the predicted correlation matrix in Figure 1(b) suggest that the correlations predicted by Transformer-based forecasters are mainly concentrated on and near the diagonal, where there is a substantial portion of the low-frequency characteristics. Based on the above observations, we can see that Transformer-based models often prioritize low-frequency features over high-frequency features, raising the possibility of frequency bias and information loss in time series forecasting.

In this work, we aim to capture informative and fine-grained dependencies to avoid frequency bias in Transformer architectures for accurate multivariate time series forecasting. To be concrete, we first establish a rigorous analysis of the feature degeneration in Transformer-based models for multivariate time series forecasting, which provides the theoretical basis for our practice. Secondly, we propose a **f**requency-**a**ware **d**ebiasing Trans**former** called **FADformer**, where i) the AttnDeb module to address the limitation of the low-pass filtering nature in the self-attention mechanism; ii) the FeatDeb module to capture inherent patterns and periodicity by amplifying the high-frequency signals. From a correlation modeling perspective, AttnDeb employs frequency domain decomposition to dynamically assign new weights to the high-frequency components of the attention maps. From a feature scaling perspective, FeatDeb extracts high-frequency information into the residual connection terms to compensate for the important patterns that are overlooked by feature degeneration. Both AttnDeb and FeatDeb are highly efficient in terms of memory usage and computational cost, which facilitates their generalization as plugins to different Transformer frameworks.

Our contributions can be summarized as follows:

- By visualizing the frequency spectrum bias in time series and correlations, we establish the first rigorous analysis of Transformer-based forecasters to reveal the oversmoothing issue, which provides theoretical support for our practical implementation.

- We propose **FADformer**, a **f**requency-**a**ware **d**ebiasing Trans**former**, which designs two efficient modules (i.e., AttnDeb and FeatDeb) to integrate low- and high-frequency components of the attention maps and feature maps for accurate forecasting.

- We show the superiority of FADformer across 13 benchmarks, which includes the best forecasting performance among 8 state-of-the-art baselines and a good generalization to existing Transformer-based methods.

## 2 RELATED WORK

**Transformer for Multivariate Time Series Forecasting.** Going beyond contemporaneous RNNs (Du et al., 2021; Lin et al., 2023), TCNs (Luo & Wang, 2024; Chen et al., 2020), and MLPs , Transformers have exhibited powerful temporal dynamics and inter-variable correlations modeling capability, leading to the trend of passionate modifications adapted for time series forecasting. Going beyond contemporaneous RNNs (Du et al., 2021; Lin et al., 2023), TCNs (Luo & Wang, 2024; Chen et al., 2020), and MLPs (Zeng et al., 2023; Han et al., 2024), Transformers have exhibited powerful temporal dynamics and inter-variable correlations modeling capability, leading to the trend of passionate modifications adapted for time series forecasting. Earlier attempts (Zhou et al., 2021; Liu et al., 2022b) have focused on temporal dependency modeling and complexity optimization for long sequences. Soon afterward, various sophisticated designs, such as Decomposition (Wu et al., 2021), Stationarization (Liu et al., 2022c), Channel Independence (Dai et al., 2024), and Patching (Nie et al., 2023), which are used to better capture temporal dependencies, have brought about consistently improved performance for Transformers. In addition, the success of Crossformer (Zhang & Yan, 2023) and iTransformer (Liu et al., 2024) has provided considerable inspiration for this domain, particularly in channel-aware multivariate correlation modeling. For instance, Fredformer (Piao et al., 2024) introduces a frequency-based channel-wise attention mechanism to mitigate frequency bias for accurate forecasting. More recently, refurbishing the Transformer in both aspects of temporal patterns and multivariate correlations, Leddam (Yu et al., 2024) employs the dual attention module to model inter-series dependencies and intra-series variations, while DUET (Qiu et al., 2024b) designs a dual clustering module to capture fine-grained distributions and relationships. Unlike the above methods, this paper proposes a frequency-aware debiasing framework to mitigate representation degeneration and performance degradation in Transformer-based forecasters.

**Oversmoothing in Transformer.** Oversmoothing was first introduced in the context of graph convolutional networks (GCNs) research (Li et al., 2018), where node features tend to become exponentially trapped in the nullspace of the graph Laplacian matrix. Coincidentally, some oversmoothing-like phenomena (Tang et al., 2021; Yan et al., 2022) occurred in Transformers, such as attention collapse, patch or token uniformity, and dimensional collapse. To mitigate oversmoothing, several approaches have been proposed for various domains. In computer vision, (Wang et al., 2022) introduces two Fourier-domain scaling techniques to address the low-pass filtering effect of the self-attention mechanism, while (Guo et al., 2023) develops a novel normalization layer inspired by contrastive learning to alleviate dimensional collapse. Meanwhile, (Fan et al., 2023) and (Shin et al., 2024) sequentially employ one smoothing regularization and an attentive inductive bias to strengthen representation diversity in sequential recommendation. In this paper, we rethink the oversmoothing issue from both the low-pass filtering nature of the self-attention mechanism and representation degeneration for multivariate time series forecasting, which, to the best of our knowledge, is the first attempt to improve forecasting performance in terms of anti-oversmoothing.

## 3 METHODOLOGY

**Definition 3.1** (Time Series Forecasting). *A multivariate time series $\mathbf{X} \in \mathbb{R}^{N \times T}$ consists of $N$ channels or variates, where each channel or variate over a look-back window of length $T$. The forecasting task is to predict the future data $\hat{\mathbf{X}} \in \mathbb{R}^{N \times H}$ for next $H$ time steps with a mapping function $\mathcal{F}$:*

$$\hat{\mathbf{X}}^{N \times H} = \mathcal{F}(\mathbf{X}^{N \times T}). \tag{1}$$

Based on Definition 3.1, our goal is to explore one more efficient Transformer-based mapping function for multivariate time series forecasting.

### 3.1 THEORETICAL ANALYSIS OF TRANSFORMER IN MTSF

The main analytical foundations focus on the feature degradation caused by the attention mechanism's oversmoothing effect and its solution via effective rank optimization. As the key ingredient of Transformers, the self-attention mechanism (Vaswani et al., 2017) can be formulated as:

$$Attn(\mathbf{X}) = softmax \left( \frac{\mathbf{Q}(\mathbf{X})(\mathbf{K}(\mathbf{X}))^{\top}}{\sqrt{d}} \right) \mathbf{V}(\mathbf{X}), \tag{2}$$

where $\mathbf{Q}(\mathbf{X}) = \mathbf{X}\mathbf{W}_Q$, $\mathbf{K}(\mathbf{X}) = \mathbf{X}\mathbf{W}_K$, $\mathbf{V}(\mathbf{X}) = \mathbf{X}\mathbf{W}_V$, $d$ is the scale factor.

**Theorem 3.2** (Oversmoothing Effect of Self-Attention). *Let $\mathbf{A} = softmax(\mathbf{O})$ be the attention matrix where $\mathbf{O} \in \mathbb{R}^{m \times m}$. For any input $\mathbf{X} \in \mathbb{R}^{m \times d}$, the attention output satisfies:*

$$\|Attn(\mathbf{X}) - \mathbf{1}\mu^\top\|_F \leq \rho(\mathbf{A})\|\mathbf{X} - \mathbf{1}\mu^\top\|_F, \tag{3}$$

*where $\mu$ is the mean vector and $\rho(\mathbf{A}) < 1$ is the contraction coefficient. This suggests that repeated application of attention results in oversmoothing.*

*Proof.* Since $\mathbf{A}$ is a row-stochastic matrix, by Perron-Frobenius theorem (Meyer, 2000; He & Wai, 2021), it has eigenvalue 1 with corresponding eigenvector $\mathbf{1}$. For any input $\mathbf{X}$, we have:

$$\|Attn(\mathbf{X}) - \mathbf{1}\mu^\top\|_F = \|\mathbf{A}\mathbf{X}\mathbf{W}_V - \mathbf{1}\mu^\top\|_F$$
$$\leq \|\mathbf{A}\|_2\|\mathbf{X}\mathbf{W}_V - \mathbf{1}\mu^\top\|_F$$
$$\leq \rho(\mathbf{A})\|\mathbf{X} - \mathbf{1}\mu^\top\|_F,$$

where $\rho(\mathbf{A})$ is the second largest eigenvalue of $\mathbf{A}$, satisfying $\rho(\mathbf{A}) < 1$. $\square$

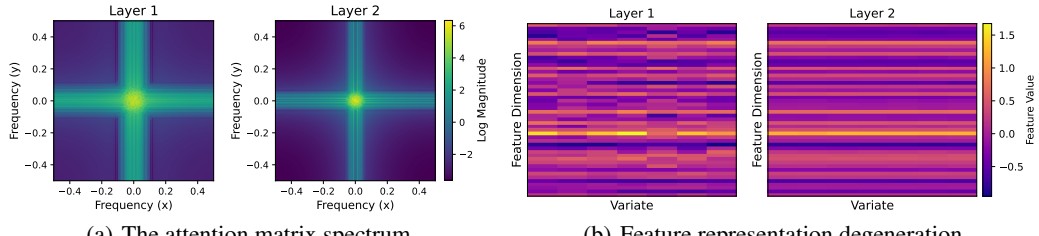

| (a) The attention matrix spectrum | (b) Feature representation degeneration |

Figure 2: Visualization of the attention matrix spectrum and feature representation degeneration sampled from the ECL datasets.

The oversmoothing effect characterized in Theorem 3.2 directly reveals the low-pass filter nature of the attention mechanism and impacts the expressive capability of features. As shown in Figure 2, the attention spectrum (i.e., Figure 2(a)) and feature maps (i.e., Figure 2(b)) from deeper layers exhibit reduced variation, indicating loss of crucial information for capturing diverse temporal patterns.

**Definition 3.3** (**Effective Rank**). *Considering matrix $\mathbf{X} \in \mathbb{R}^{n \times d}$, whose singular value decomposition is given by $\mathbf{X} = \mathbf{U}\mathbf{Q}\mathbf{V}$, where $\mathbf{Q}$ is a diagonal matrix with singular value $\lambda_1 \geq \lambda_2 \geq \cdots \geq \lambda_r \geq 0$ with $r = \min\{n, d\}$. The distribution of singular values is defined as $L_1$-normalized form $q_i = \lambda_i / \sum_{j=1}^r |\lambda_j|$. The effective rank of the matrix $\mathbf{X}$, can be formulated as $\mathcal{E}(\mathbf{X}) = \exp\left(H(q_1, \cdots, q_r)\right)$, where $H(q_1, \cdots, q_r) = -\sum_{j=1}^r q_j \log q_j$ is the Shannon entropy.*

To address the feature degeneration, we introduce the effective rank (Roy & Vetterli, 2007; Guo et al., 2023) as a measure of feature diversity, where higher values indicate more balanced utilization of feature dimensions while lower values suggest concentration on a few dominant directions. From a global perspective, a full attention block of the Transformer includes the self-attention mechanism and residual connection, which can be formulated as:

$$\mathbf{X}' = (1+s) \cdot \mathbf{X} - s \cdot \left(\mathbf{I} - softmax(\mathbf{X}\mathbf{X}^\top)\right)\mathbf{X}, \tag{4}$$

where $s \in (0, 1)$ is a scale factor. When $softmax(\mathbf{Q}(\mathbf{X})(\mathbf{K}(\mathbf{X}))^\top) = \mathbf{I}$, we can obtain the upper boundary $\mathbf{X}'_u \leq (1+s)\mathbf{X}$. In this way, we can adjust the effective rank of the features by using $s$.

**Proposition 3.4.** *Considering the updated form $\mathbf{X}' = (1+s)\mathbf{X} - s(\mathbf{X}\mathbf{X}^\top)\mathbf{X}$, let $\lambda_{max}$ be the largest singular value of $\mathbf{X}$. For $s > 0$ satisfying $1 + (1 - \lambda_{max}^2)s > 0$, we have $\mathcal{E}(\mathbf{X}') > \mathcal{E}(\mathbf{X})$.*

Proposition 3.4 gives a theoretical possibility of alleviating representation degeneration by utilizing the effective rank. When $s$ satisfies $1 + (1 - \lambda_{max}^2)s > 0$, this update can increase the effective rank for the representation matrix. Note that, whether $|\lambda_{max}| > 1$ or not, the condition will be satisfied if $s$ is sufficiently close to 0. By maximizing the effective rank of feature representations, we can mitigate the oversmoothing effect and improve feature diversity. This is particularly crucial for multivariate time series forecasting, where capturing diverse temporal patterns across multiple variates requires rich feature representations. Please see Appendix A and B for more details.

## 3.2 FREQUENCY-AWARE DEBIASING FRAMEWORK

To address the oversmoothing problem in MTSF, we propose FADformer, a novel frequency-aware debiasing framework for Transformer-based models. Next, we will elaborate on the architecture and implementation of FADformer. Considering that Transformer-based models can model inherent dependencies of time series in both the time and variate dimensions, we take iTansformer (Liu et al., 2024) as the representative backbone for convenience.

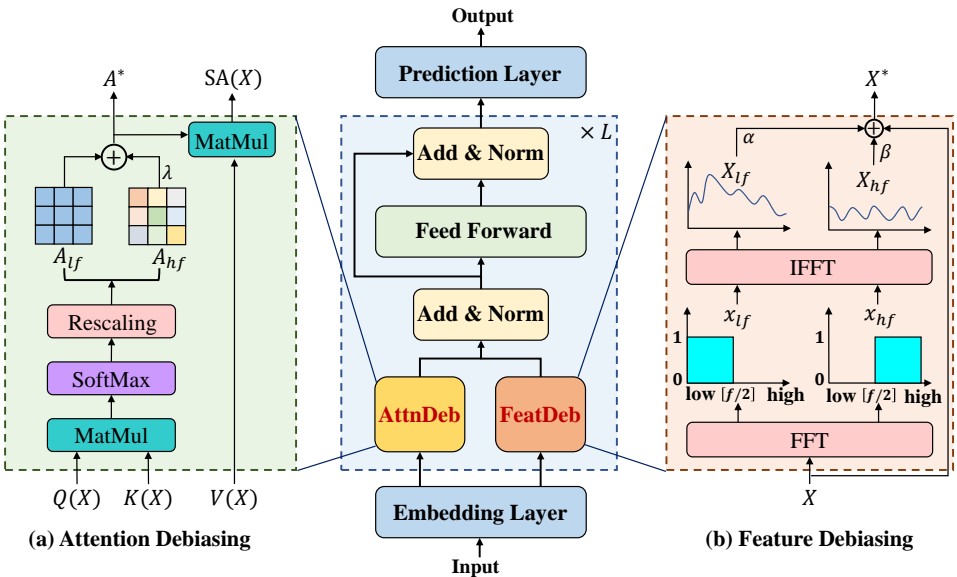

Figure 3: Architecture of our proposed FADformer. We design two plugin modules (i.e., AttnDeb and FeatDeb) based on the self-attention mechanism, which can dynamically assign weights to high-frequency components of the self-attention and feature maps via Fourier transformation.

**Overall Structure.** Figure 3 shows the architecture of our FADformer, which adopts a dual frequency-aware debiasing strategy on both attention and feature spaces, simultaneously capturing more fine-grained dependencies for accurate multivariate time series forecasting. Concretely, we first use the Instance Norm to unify the distribution of the model input $\mathbf{X} \in \mathbb{R}^{N \times T}$. Then, we feed $\mathbf{X}$ to the embedding layer, which transforms $\mathbf{X}$ into feature embeddings $\mathbf{F} \in \mathbb{R}^{N \times d}$ from the channel dimension. Next, the AttnDeb module utilizes an implementation-friendly decomposition mechanism to reorganize the attention components. Meanwhile, the FeatDeb module reweights the low-frequency and high-frequency components of feature representations from the Fourier spectral domain. After replacing the original residual connection of iTransformer with the FeatDeb, we perform the Addition and Layer Norm operations to the outputs of the AttnDeb and FeatDeb modules. Finally, following the structure of iTransformer, we successively add the Feed-Forward Network (FFN), Add-Norm Layer, and Prediction Layer to the model for ultimate forecasting.

**Attention Debiasing.** As mentioned in Section 3.1, the self-attention mechanism is limited to performing low-pass filtering, which hinders the expressiveness of Transformer-based forecasters in the filter space. Inspired by trend-seasonal decomposition (Wu et al., 2021), we design a scaling technique that directly manipulates the attention matrix, termed AttnDeb, to reweight the contributions of low-pass and high-pass filtering, thereby enabling the creation of all-pass filters.

Formally, let $\mathbf{A}$ denote a self-attention matrix, $\tilde{\mathbf{A}} = \mathcal{F}\mathbf{A}\mathcal{F}^{-1}$ be the spectral response of $\mathbf{A}$. Then, we can extract the low-pass spectral response from $\mathbf{A}$ as:

$$\mathcal{LF}(\mathbf{A}) = \mathcal{F}^{-1}\phi(\mathbf{A})\mathcal{F}. \tag{5}$$

To this end, we must explore an appropriate function $\phi(\mathbf{A})$ to satisfy the low-pass filtering property. Considering the smoothness and low-pass properties of the Gaussian distribution (Conejo et al., 2022), we construct a Gaussian kernel function to obtain the low-pass component of $\mathbf{A}$. Concretely, we take the number of variates $N$ as the index length $\mathcal{I} = \{0, 1, \cdots, N-1\}$ and compute the distance matrix between the indexes $M^D_{(i,j)} = |i-j|, i, j \in \mathcal{I}$. Regarding $\sqrt{N}$ as the standard

deviation of the Gaussian distribution, our distance-based attention weight matrix is formulated as:

$$Attn^D_{(i,j)} = \exp\left(-\frac{-(i-j)^2}{2N}\right). \tag{6}$$

Afterward, our $\phi(\mathbf{A})$ can be written as a normalized attention weight matrix to ensure that the sum of the weights in each row is equal to 1:

$$\phi(\mathbf{A}) = \frac{\exp\left(-\frac{(i-j)^2}{2N}\right)}{\sum_{m=0}^{N-1}\exp\left(-\frac{-(i-m)^2}{2N}\right)}. \tag{7}$$

Accordingly, we take the complementary part of $\phi(\mathbf{A})$ as the high-pass filter of $\mathbf{A}$, i.e., $\psi(\mathbf{A}) = \mathbf{A} - \phi(\mathbf{A})$. To avoid high-frequency components being filtered out, we introduce a trainable parameter $\lambda$ to rescale $\psi(\mathbf{A})$ and match the magnitude of $\phi(\mathbf{A})$. In this way, we can obtain a recomputed attention map $\mathbf{X}'$ as follows:

$$\mathbf{A}' = \phi(\mathbf{A}) + (1+\lambda)\psi(\mathbf{A}). \tag{8}$$

Note that $\lambda$ is related to the number of attention heads, and we omit the number of Transformer layers for convenience.

**Feature Debiasing.** According to our analysis in Section 3.1, encoded features suffer from the representation degeneration issue even at shallow layers, due to the inherently smooth nature of time series. Recalling Equation (4), we can alleviate the representation degeneration by Proposition 3.4. Considering the item $s\mathbf{X}$ of the Equation (4) as a complement $r(\mathbf{X})$ to the input $\mathbf{X}$, then the attention block with the residual connection can be written as:

$$\mathbf{X}^* = \mathbf{X} + r(\mathbf{X}) + softmax(\mathbf{Q}(\mathbf{X})\mathbf{K}(\mathbf{X})^\top)\mathbf{V}(\mathbf{X}). \tag{9}$$

Removing the transformation functions of $\mathbf{Q}(\cdot), \mathbf{K}(\cdot)$, and $\mathbf{V}(\cdot)$, we can simplify the above formula to $\mathbf{X}^* = (1+s)\mathbf{X} + softmax(\mathbf{X}\mathbf{X}^\top)\mathbf{X}$. Based on Proposition 3.4, we can quickly obtain that $\mathcal{E}(\mathbf{X}^*) > \mathcal{E}(\mathbf{X})$. In this way, we can rescale the input $\mathbf{X}$ in the residual connection to avoid representation degeneration.

To address the feature degradation that occurs as layers increase, we propose an alternative scaling technique, named FeatDeb, which reweights the feature maps of the residual connection using the Fourier transformation. Specifically, we first utilize the Discrete Fourier Transform $DFT(\cdot)$ to transform feature maps $\mathbf{X} \in \mathbb{C}^{N \times d}$ into frequency components $\mathbf{Z} \in \mathbb{C}^{N \times d}$. Then, we select the frequency set with the top $K$ largest amplitude as $TopK(f_1, f_2, \cdots, f_N)$, where $f$ denotes the Fourier basis. The low-frequency information of feature maps can be formulated with $IDFT(\cdot)$ as follows:

$$\begin{aligned}\mathbf{Z} &= DFT(\mathbf{X}), \\ \mathbf{X}_{LF} &= IDFT(TopK(f_1, \cdots, f_N)\mathbf{Z}).\end{aligned} \tag{10}$$

With $\mathbf{X}_{LP}$, we can easily get the high-frequency information from $\mathbf{X}$ by removing $\mathbf{X}_{LP}$:

$$\mathbf{X}_{HF} = \mathbf{X} - \mathbf{X}_{LF}. \tag{11}$$

To balance the weights of decomposed feature maps, we introduce two group-trainable parameters $\alpha, \beta \in \mathbb{R}^d$ to rescale different components. Therefore, as an equivalent substitution for $s\mathbf{X}$, $r(\mathbf{X})$ can be written as:

$$r(\mathbf{X}) = \alpha\mathbf{X}_{LF} + \beta\mathbf{X}_{HF}. \tag{12}$$

Finally, our FeatDeb in the residual connection can be formulated as follows:

$$\mathbf{X}^* = \mathbf{X} + \alpha\mathbf{X}_{LF} + \beta\mathbf{X}_{HF}. \tag{13}$$

## 4 EXPERIMENTS

We conduct extensive experiments to evaluate the performance of FADformer, covering 13 real-world datasets and 8 well-acknowledged baselines. All the experiments are done on a single NVIDIA 4090 24GB GPU using the Pytorch (Paszke et al., 2019) framework.

Table 1: Multivariate forecasting performance. The lookback length is set to $T = 96$ and all the results are averaged from all predictions $H \in \{12, 24, 48, 96\}$ for PEMS and $H \in \{96, 192, 336, 720\}$ for other benchmarks. See Table 12 in Appendix H for the full results.

| Models | FADformer (Ours) | | SOFTS (2024) | | iTransformer (2024) | | PatchTST (2023) | | Crossformer (2023) | | DLinear (2023) | | TimesNet (2023) | | SCINet (2022) | | FEDformer (2022) | |
|---|---|---|---|---|---|---|---|---|---|---|---|---|---|---|---|---|---|---|
| Metric | MSE | MAE | MSE | MAE | MSE | MAE | MSE | MAE | MSE | MAE | MSE | MAE | MSE | MAE | MSE | MAE | MSE | MAE |
| ETTh1 | 0.443 | 0.434 | 0.449 | 0.443 | 0.454 | 0.448 | 0.453 | 0.446 | 0.529 | 0.522 | 0.456 | 0.452 | 0.458 | 0.450 | 0.747 | 0.647 | 0.440 | 0.460 |
| ETTh2 | 0.376 | 0.399 | 0.377 | 0.400 | 0.383 | 0.407 | 0.385 | 0.410 | 0.942 | 0.684 | 0.599 | 0.515 | 0.414 | 0.427 | 0.954 | 0.723 | 0.437 | 0.449 |
| ETTm1 | 0.390 | 0.390 | 0.395 | 0.402 | 0.407 | 0.410 | 0.396 | 0.406 | 0.513 | 0.495 | 0.403 | 0.407 | 0.400 | 0.406 | 0.486 | 0.481 | 0.448 | 0.452 |
| ETTm2 | 0.279 | 0.319 | 0.288 | 0.330 | 0.288 | 0.332 | 0.287 | 0.330 | 0.757 | 0.611 | 0.350 | 0.401 | 0.291 | 0.333 | 0.571 | 0.537 | 0.305 | 0.349 |
| ECL | 0.166 | 0.255 | 0.174 | 0.264 | 0.178 | 0.270 | 0.189 | 0.276 | 0.244 | 0.334 | 0.212 | 0.300 | 0.193 | 0.295 | 0.268 | 0.365 | 0.214 | 0.327 |
| Traffic | 0.412 | 0.266 | 0.421 | 0.275 | 0.428 | 0.282 | 0.454 | 0.286 | 0.550 | 0.304 | 0.625 | 0.383 | 0.620 | 0.336 | 0.804 | 0.509 | 0.610 | 0.376 |
| Solar | 0.229 | 0.230 | 0.230 | 0.256 | 0.233 | 0.262 | 0.236 | 0.266 | 0.641 | 0.639 | 0.330 | 0.401 | 0.301 | 0.319 | 0.282 | 0.375 | 0.261 | 0.381 |
| Weather | 0.249 | 0.267 | 0.258 | 0.279 | 0.258 | 0.278 | 0.256 | 0.279 | 0.259 | 0.315 | 0.265 | 0.317 | 0.259 | 0.287 | 0.292 | 0.363 | 0.309 | 0.360 |
| Exchange | 0.354 | 0.401 | 0.374 | 0.412 | 0.360 | 0.403 | 0.367 | 0.404 | 0.940 | 0.707 | 0.354 | 0.414 | 0.416 | 0.443 | 0.750 | 0.626 | 0.519 | 0.429 |
| PEMS03 | 0.108 | 0.214 | 0.112 | 0.219 | 0.116 | 0.226 | 0.180 | 0.291 | 0.169 | 0.282 | 0.278 | 0.375 | 0.147 | 0.248 | 0.114 | 0.224 | 0.213 | 0.327 |
| PEMS04 | 0.108 | 0.216 | 0.114 | 0.226 | 0.121 | 0.232 | 0.195 | 0.307 | 0.209 | 0.314 | 0.295 | 0.388 | 0.129 | 0.241 | 0.093 | 0.202 | 0.231 | 0.337 |
| PEMS07 | 0.102 | 0.195 | 0.100 | 0.203 | 0.100 | 0.204 | 0.211 | 0.303 | 0.235 | 0.315 | 0.329 | 0.396 | 0.125 | 0.226 | 0.119 | 0.234 | 0.165 | 0.283 |
| PEMS08 | 0.147 | 0.226 | 0.148 | 0.234 | 0.151 | 0.234 | 0.280 | 0.321 | 0.268 | 0.307 | 0.379 | 0.416 | 0.193 | 0.271 | 0.159 | 0.244 | 0.286 | 0.358 |
| 1st Count | 10 | 12 | 1 | 0 | 1 | 0 | 0 | 0 | 0 | 0 | 1 | 0 | 0 | 0 | 1 | 1 | 1 | 0 |

## 4.1 EXPERIMENTAL SETUP

**Datasets.** We use 13 challenging datasets in multivariate time series forecasting as datasets, including 4 ETT (Zhou et al., 2021) subsets (ETTh1, ETTh2, ETTm1, and ETTm2), Electricity (Wu et al., 2021), Traffic (Wu et al., 2021), Exchange (Wu et al., 2021), Weather (Wu et al., 2021), Solar-Energy (Lai et al., 2018), and four PEMS (Liu et al., 2024) subsets (PEMS03, PEMS04, PEMS07, and PEMS08). Please refer to Appendix C for detailed descriptions of all the datasets.

**Evaluation.** Following iTransformer (Liu et al., 2024), we use Mean Squared Error (MSE) and Mean Absolute Error (MAE) as the core metrics for the evaluation. For a fair comparison of forecasting performance, we adopt a consistent protocol to set the historical horizon length to $T = 96$ and prediction lengths $H \in \{12, 24, 48, 96\}$ for PEMS and $H \in \{96, 192, 336, 720\}$ for other datasets. Additionally, the number of training epochs is fixed at 10 for all models. Detailed hyperparameters can be found in Appendix C.

**Baselines.** We compare FADformer with 8 representative baselines, including 1) Transformer-based iTransformer (Liu et al., 2024), PatchTST (Nie et al., 2023), Crossformer (Zhang & Yan, 2023), FEDformer (Zhou et al., 2022); 2) Linear-based SOFTS (Han et al., 2024), DLinear (Zeng et al., 2023); and 3) TCN-based TimesNet (Wu et al., 2023a), SCINet (Liu et al., 2022a).

## 4.2 MAIN RESULTS

**Forecasting performance.** As shown in Table 1, FADformer achieves the best or second-best forecasting outcomes in all 13 benchmarks on the averaging prediction horizon. Specifically, FADformer ranks in the top 1 of MSE and MAE among 10 and 12 out of 13 benchmarks, respectively. Furthermore, compared to previous state-of-the-art methods, FADformer shows significant advancements. For instance, our method reduces the average MSE error from 0.174 to 0.166 on ECL, representing a notable 4.5% reduction. For challenging datasets like Solar and Weather, FADformer achieves a substantial relative decrease of 10.2% and 4.0% in average MAE error, respectively. These significant improvements indicate that our FADformer possesses robust performance and broad applicability in multivariate time series forecasting tasks.

**Generalization performance.** To evaluate the generalization of FADformer, we apply our framework to three well-established Transformer-based models, i.e., TimeBridge (Liu et al., 2025), TQNet (Lin et al., 2025), and Leddam (Yu et al., 2024), which model the inherent dependencies of multivariate time series with well-designed architectures. As shown in Table 2, we integrate our debiasing modules into three representative Transformer-based models without modifying the original structures and their hyperparameters. Compared to the baselines, our debiasing modules, as plugins, can achieve substantial performance improvements across Weather, Solar, ECL, and Traffic datasets, which contain time values of 21, 137, 321, and 862 variates, respectively. In particular, we outper-

Table 2: Generalization performance across four datasets. The lookback length matches that of the baseline models, and all the results are averaged from four prediction horizons $H \in \{96, 192, 336, 720\}$. See Table 9 in Appendix E for the full results.

| Models | TimeBridge (2025) | | + Debiasing | | TQNet (2025) | | + Debiasing | | Leddam (2024) | | + Debiasing | |
|---|---|---|---|---|---|---|---|---|---|---|---|---|
| | MSE | MAE | MSE | MAE | MSE | MAE | MSE | MAE | MSE | MAE | MSE | MAE |
| Weather | 0.219 | 0.249 | **0.216** | **0.248** | 0.242 | 0.269 | **0.239** | **0.267** | 0.242 | 0.272 | **0.238** | **0.267** |
| Solar | 0.181 | 0.239 | **0.179** | **0.238** | 0.198 | 0.256 | **0.196** | **0.254** | 0.230 | 0.264 | **0.228** | **0.261** |
| ECL | 0.149 | 0.245 | **0.147** | **0.244** | 0.164 | 0.259 | **0.162** | **0.257** | 0.169 | 0.263 | **0.166** | **0.258** |
| Traffic | 0.360 | 0.255 | **0.357** | **0.253** | 0.445 | 0.276 | **0.432** | **0.271** | 0.467 | 0.294 | **0.438** | **0.288** |

form TimeBridge by a $1.4\%$ MSE reduction in Weather, achieve lower MSE scores than TQNet by a $1.2\%$ reduction in ECL, and decrease the MSE score of Leddam by $6.2\%$ in Traffic. These improvements demonstrate that our debiasing modules are compatible with various Transformer-based forecasters. Moreover, we explore the generalization of FADformer under different attention mechanisms, which is provided in Appendix E.

Table 3: Ablation study for debiasing modules across six datasets with the input length $T = 96$ and the prediction horizon $H = 96$.

| Models | ETTm1 | | ECL | | Traffic | | Weather | | Solar | | PEMS03 | |
|---|---|---|---|---|---|---|---|---|---|---|---|---|
| | MSE | MAE | MSE | MAE | MSE | MAE | MSE | MAE | MSE | MAE | MSE | MAE |
| iTransformer | 0.334 | 0.368 | 0.148 | 0.240 | 0.395 | 0.268 | 0.174 | 0.214 | 0.203 | 0.237 | 0.168 | 0.279 |
| w/o AttnDeb | 0.325 | 0.353 | 0.144 | 0.235 | 0.389 | 0.259 | 0.169 | 0.207 | 0.201 | 0.212 | 0.160 | 0.271 |
| w/o FeatDeb | 0.322 | 0.354 | 0.146 | 0.232 | 0.390 | 0.261 | 0.166 | 0.204 | 0.201 | 0.221 | 0.159 | 0.269 |
| **FADformer (Ours)** | **0.319** | **0.349** | **0.141** | **0.229** | **0.386** | **0.254** | **0.162** | **0.196** | **0.199** | **0.208** | **0.155** | **0.264** |

## 4.3 ABLATION STUDIES

**Effectiveness of debiasing modules.** To verify the effectiveness of our FADformer, we provide an indispensable ablation study with each debiasing module. To be concrete, we employ iTransformer (Liu et al., 2024) as the baseline, and disable AttenDeb or FeatDeb as model variants on six datasets. As shown in Table 3, we observe significant performance improvements across all six datasets. For instance, the FeatDeb module achieves an average of $4.4\%$ decrease in MAE across six datasets compared to iTransformer, while the AttenDeb module achieves a $4.1\%$ improvement in MAE. Moreover, their synergistic integration yields further enhancements in forecasting performance, resulting in an average $6.8\%$ decrease in MAE, reaching an optimal level. These results indicate that oversmoothing in attention-based forecasters limits the multivariate time series forecasting performance, supporting the efficacy and rationality of both proposed debiasing modules.

**Variants of debiasing modules.** The comparison of different debiasing strategies in our debiasing modules is shown in Table 4. For the AttnDeb module, **Mean** means that we compute the average value of attention weights as low-pass components, **Uniform** refers to capturing the low-pass component of attention weights with uniform distribution, and **Gaussian** learns the low-pass weights by a Gaussian distribution. For the FeatDeb module, **Mean** indicates that the low-frequency components are obtained by the average

Table 4: Ablation for different debiasing variants.

| Variants | | AttnDeb | | | FeatDeb | | |
|---|---|---|---|---|---|---|---|
| | | Mean | Uniform | Gaussian | Mean | First-K | Top-K |
| Weather | MSE | 0.255 | 0.254 | **0.249** | 0.254 | 0.255 | **0.249** |
| | MAE | 0.272 | 0.270 | **0.267** | 0.273 | 0.276 | **0.267** |
| Solar | MSE | 0.233 | 0.234 | **0.229** | 0.233 | 0.235 | **0.229** |
| | MAE | 0.242 | 0.234 | **0.230** | 0.239 | 0.242 | **0.230** |
| ECL | MSE | 0.170 | 0.170 | **0.166** | 0.171 | 0.170 | **0.166** |
| | MAE | 0.256 | 0.258 | **0.255** | 0.263 | 0.259 | **0.255** |
| Traffic | MSE | 0.420 | 0.426 | **0.412** | 0.419 | 0.417 | **0.412** |
| | MAE | 0.277 | 0.274 | **0.266** | 0.276 | 0.273 | **0.266** |

value of all the series representations, **First-K** defines the first $K$ smallest frequency indices in the Fourier transform as the low-frequency components of features, and **Top-K** selects the frequency set with the top $K$ largest amplitude as low-frequency components. Note that our default AttnDeb and FeatDeb are **Gaussian** and **Top-K** strategies, respectively. From Table 5, we can observe that the Gaussian and Top-K strategies bring lower MSE and MAE results than other variants, which reveals that both strategies are the optimal choice for our AttnDeb and FeatDeb modules, respectively.

## 4.4 MODEL ANALYSIS

**Efficiency analysis.** Our FADformer is a Transformer-based architecture with two plug-in debiasing modules, which bring negligible extra computational overhead to the model. Specifically, our AttnDeb with the Uniform strategy introduces only $O(hL)$ extra parameters, where $h$ is the number of heads and $L$ is the number of Transformer blocks. As for the FeatDeb, it can be implemented without an explicit Fourier transform, which is as simple as running with the mean and subtraction operations. Therefore, the computational complexity of FADformer remains $O(N^2)$ or $O(L^2)$, consistent with other Transformer architectures, since it is governed by the attention matrix calculation in the channel or time dimension. Table 6 shows the memory and time usage across representative Transformer-based forecasters on the Traffic dataset, which reveals that our debiasing strategies are efficient.

Table 5: Efficiency analysis for FADformer.

| Models | Training Times (Seconds/Epoch) | Inference Times (Seconds) | GPU (GB) | Parameter (MB) | FLOPs (GB) |
|---|---|---|---|---|---|
| Crossformer | 283.47 | 66.19 | 11.77 | 46.90 | 229.09 |
| PatchTST | 175.36 | 22.45 | 9.95 | 13.58 | 33.52 |
| iTransformer | 52.60 | 14.28 | 7.50 | 6.11 | 8.63 |
| **FADformer** | 57.39 | 14.72 | 6.64 | 5.36 | 6.49 |
| **AttnDeb** | 9.55 | 0.42 | 1.55 | 0.02 | 0.00 |
| **FeatDeb** | 5.93 | 0.01 | 0.16 | 0.00 | 0.00 |

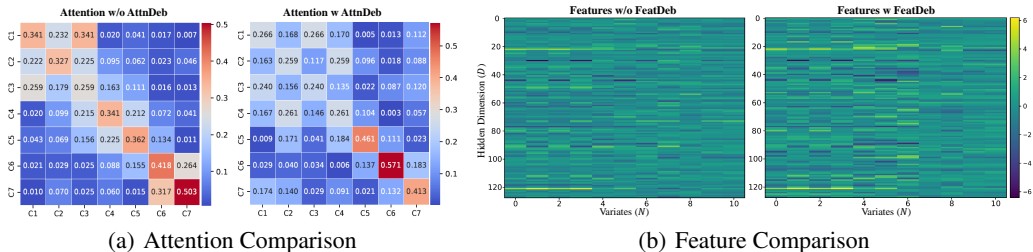

| (a) Attention Comparison | (b) Feature Comparison |
|---|---|

Figure 4: Visual attention and feature map results of different channels (i.e., C1-C7) on ETTm2. Refer to Appendix F for more details.

**Visualization analysis.** As seen in Figure 4, the attention w/o AttnDeb is mainly focused on the diagonal, while that with At-tnDeb is not, which indicates that our At-tenDeb can effectively preserve the high-frequency information. Furthermore, the feature maps obtained with FeatDeb exhibit richer and clearer color variations, indicating that our FeatDeb can alleviate feature degeneration.

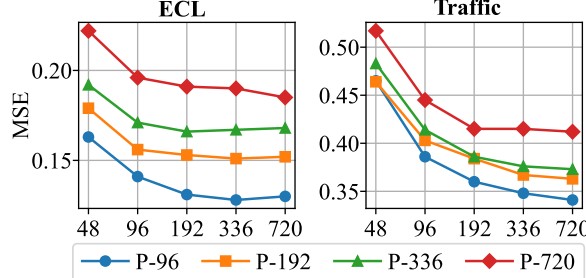

Figure 5: Lookback length analysis.

**Lookback length sensitivity.** Generally, a longer lookback window may lead to a curse of dimensionality, which hinders the model's forecasting performance. However, as seen in Figure 5, our FADformer reduces the MSE scores with enlarged historical information $T = \{48, 96, 192, 336, 720\}$ in most prediction horizons $P = \{96, 192, 336, 720\}$, which confirms that our debiasing modules can capture more informative patterns from a longer lookback window.

## 5 CONCLUSION AND FUTURE WORK

In this paper, we propose a general Transformer-based framework, FADformer, which designs a dual debiasing strategy on the attention mechanism and residual connection to enhance MTSF. Based on a rigorous theoretical analysis of the oversmoothing between correlations and features, we rescale the low- and high-frequency components of the attention maps and feature maps in the Fourier domain. These innovative techniques, albeit simple, collectively empower FADformer to achieve outstanding prediction and generalization performance. In the future, we plan to integrate these debiasing strategies into large-scale pre-trained models and to develop more general debiasing strategies for other time series analysis tasks.

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

## A    FULL PROOF THEOREM 3.2

**Theorem A.1** (Oversmoothing Effect of Self-Attention). *Let $\mathbf{A} = softmax(\mathbf{O})$ be the attention matrix where $\mathbf{O} \in \mathbb{R}^{m \times m}$ is such that $\mathbf{A}$ is irreducible and aperiodic. For any input $\mathbf{X} \in \mathbb{R}^{m \times d}$ and assuming $\mathbf{W}_V$ is a contraction (i.e., $\|\mathbf{W}_V\|_2 \leq 1$), the attention output satisfies:*

$$\|Attn(\mathbf{X}) - \mathbf{1}\mu^\top\|_F \leq \rho(\mathbf{A})\|\mathbf{X} - \mathbf{1}\mu^\top\|_F, \tag{14}$$

*where $\mu = \frac{1}{m}\mathbf{1}^\top\mathbf{X}$ is the mean vector and $\rho(\mathbf{A}) < 1$ is the second largest eigenvalue modulus of $\mathbf{A}$. This suggests that repeated application of attention results in oversmoothing.*

*Proof.* Since $\mathbf{A}$ is a row-stochastic matrix (by the softmax construction), by the Perron-Frobenius theorem (Meyer, 2000; He & Wai, 2021), it has eigenvalue 1 with corresponding right eigenvector $\mathbf{1}$ and left eigenvector $\boldsymbol{\pi}^\top$ (the stationary distribution).

For irreducible and aperiodic stochastic matrices, we have $\rho(\mathbf{A}) < 1$, where $\rho(\mathbf{A})$ is the second largest eigenvalue modulus.

Let $\mu = \frac{1}{m}\mathbf{1}^\top\mathbf{X}$ be the mean vector. Note that:

$$\mathbf{A}\mathbf{1}\mu^\top = \mathbf{1}\mu^\top \quad (\text{since } \mathbf{A}\mathbf{1} = \mathbf{1})$$
$$\mathbf{1}\mu^\top\mathbf{W}_V = \mathbf{1}(\mu^\top\mathbf{W}_V)$$

Now consider:

$$\begin{aligned}
\|Attn(\mathbf{X}) - \mathbf{1}\mu^\top\mathbf{W}_V\|_F &= \|\mathbf{A}\mathbf{X}\mathbf{W}_V - \mathbf{1}\mu^\top\mathbf{W}_V\|_F \\
&= \|\mathbf{A}(\mathbf{X} - \mathbf{1}\mu^\top)\mathbf{W}_V + (\mathbf{A}\mathbf{1}\mu^\top - \mathbf{1}\mu^\top)\mathbf{W}_V\|_F \\
&= \|\mathbf{A}(\mathbf{X} - \mathbf{1}\mu^\top)\mathbf{W}_V\|_F \quad (\text{since } \mathbf{A}\mathbf{1}\mu^\top = \mathbf{1}\mu^\top) \\
&\leq \|\mathbf{A}\|_2\|(\mathbf{X} - \mathbf{1}\mu^\top)\mathbf{W}_V\|_F \\
&\leq \|\mathbf{A}\|_2\|\mathbf{W}_V\|_2\|\mathbf{X} - \mathbf{1}\mu^\top\|_F
\end{aligned}$$

For the spectral norm of $\mathbf{A}$, we have $\|\mathbf{A}\|_2 = 1$ since $\mathbf{A}$ is stochastic. However, we can obtain a tighter bound using the contraction property. Note that:

$$\mathbf{A}(\mathbf{X} - \mathbf{1}\mu^\top) = \mathbf{A}\mathbf{X} - \mathbf{1}\mu^\top$$

since $\mathbf{A}\mathbf{1}\mu^\top = \mathbf{1}\mu^\top$.

The matrix $\mathbf{A}$ acts as a contraction on the subspace orthogonal to $\mathbf{1}$, with contraction coefficient $\rho(\mathbf{A})$. Therefore:

$$\|\mathbf{A}(\mathbf{X} - \mathbf{1}\mu^\top)\|_F \leq \rho(\mathbf{A})\|\mathbf{X} - \mathbf{1}\mu^\top\|_F$$

Combining with the assumption that $\|\mathbf{W}_V\|_2 \leq 1$, we get:

$$\begin{aligned}
\|Attn(\mathbf{X}) - \mathbf{1}\mu^\top\mathbf{W}_V\|_F &\leq \rho(\mathbf{A})\|\mathbf{X} - \mathbf{1}\mu^\top\|_F\|\mathbf{W}_V\|_2 \\
&\leq \rho(\mathbf{A})\|\mathbf{X} - \mathbf{1}\mu^\top\|_F
\end{aligned}$$

This shows that each attention layer contracts the input towards its mean, with contraction coefficient at most $\rho(\mathbf{A}) < 1$. □

## B    PROOF OF PROPOSITION 3.4

**Lemma B.1.** *Let the eigenvalues of $\mathbf{A}\mathbf{A}^\top$ be $\alpha_1 \geq \alpha_2 \cdots \geq \alpha_n$ and the eigenvalues of $\mathbf{B}\mathbf{B}^\top$ be $\beta_1 \geq \beta_2 \geq \cdots \beta_n$. If $\alpha_i/\beta_i$ is increasing as $i$ increases, then we have $\mathcal{E}(\mathbf{A}) \geq \mathcal{E}(\mathbf{B})$.*

Lemma B.1 can be proved just by the definition 3.3 of the effective rank.

**Proposition 3.5.** Considering the updated form $\mathbf{X}' = (1 + s)\mathbf{X} - s(\mathbf{X}\mathbf{X}^\top)\mathbf{X}$, let $\lambda_{max}$ be the largest singular value of $\mathbf{X}$. For $s > 0$ satisfying $1 + (1 - \lambda_{max}^2)s > 0$, we have $\mathcal{E}(\mathbf{X}') > \mathcal{E}(\mathbf{X})$.

*Proof.* We have

$$\mathbf{X}' = ((1+s) - s\mathbf{X}\mathbf{X}^\top)\mathbf{X}. \tag{15}$$

Hereby,

$$\mathbf{X}'\mathbf{X}'^\top = s^2(\mathbf{X}\mathbf{X}^\top)^3 - 2s(1+s)(\mathbf{X}\mathbf{X}^\top)^2 + (1+s)^2(\mathbf{X}\mathbf{X}^\top) \tag{16}$$

Suppose $\lambda_1 \geq \lambda_2 \geq \cdots \lambda_n$ are the eigenvalues of $\mathbf{X}\mathbf{X}^\top$, then $\mathbf{X}'\mathbf{X}'^\top$ has the same eigenvectors as $\mathbf{X}\mathbf{X}^\top$, and its eigenvalues are $(\lambda_i s - (1+s))^2 \lambda_i$. Since $s$ satisfies $1 + (1 - \sigma_{max}^2 s) > 0$, we have $1 + s > \lambda_i s$. Therefore, $(\lambda_i s - (1+s))^2$ is increasing as $i$ increases, resulting the fact that $\mathbf{X}' > \mathbf{X}$ by using Lemma B.1. □

## C  IMPLEMENTATION DETAILS

**Benchmarks details**  We evaluate the performance of FADformer compared with various baselines on 13 well-established benchmarks [1], which are detailed in Table 7.

Table 6: Detailed descriptions of datasets. Channel denotes the number of variates in each dataset. The prediction length indicates four prediction settings. The dataset size is split into (Train, Validation, Test). Frequency denotes the sampling interval of time points.

| Benchmarks | Channels | Prediction Length | Dataset Size | Frequency | Information |
|---|---|---|---|---|---|
| ETTm1 | 7 | | (34465, 11521, 11521) | 15min | Electricity |
| ETTm2 | 7 | | (34465, 11521, 11521) | 15min | Electricity |
| ETTh1 | 7 | | (8545, 2881, 2881) | Hourly | Electricity |
| ETTh2 | 7 | | (8545, 2881, 2881) | Hourly | Electricity |
| ECL | 321 | {96, 192, 336, 720} | (18317, 2633, 5261) | Hourly | Electricity |
| Traffic | 862 | | (12185, 1757, 3509) | Hourly | Transportation |
| Exchange | 8 | | (5120, 665, 1422) | Daily | Economy |
| Weather | 21 | | (36792, 5271, 10540) | 10min | Weather |
| Solar-energy | 137 | | (36601, 5161, 10417) | 10min | Electricity |
| PEMS03 | 358 | | (15617, 5135, 5135) | 5min | Transportation |
| PEMS04 | 307 | {12, 24, 48, 96} | (10172, 3375, 3375) | 5min | Transportation |
| PEMS07 | 883 | | (16911, 5622, 5622) | 5min | Transportation |
| PEMS08 | 170 | | (10690, 3548, 3548) | 5min | Transportation |

We follow the train-validation-test set split protocol of TimesNet (Wu et al., 2023a), where each part is strictly divided according to chronological order, ensuring no data leakage issues.

**Training details.**  We trained our FADformer using the L1 loss function and employed the ADAM optimizer. We initialized the random seed as $seed = 2021$. During the training process, the number of training epochs is fixed to 10 and we carried out a grid hyperparameter search for the optimal selection. Specifically, we set hidden dimension of layer $d \in \{64, 128, 256, 512\}$, learning rate $lr \in \{0.001, 0.0005, 0.0001\}$, dropout rate $dr \in \{0.0, 0.1, 0.2, 0.5\}$, and the number of transformer blocks $L \in \{1, 2, 3, 4\}$.

**Metrics details**  Regarding evaluation metrics, we utilize the mean square error (MSE) and mean absolute error (MAE) for long-term and short-term forecasting:

$$\text{MSE} = \frac{1}{L}\sum_{i=1}^{L}(\mathbf{X}_i - \hat{\mathbf{X}}_i)^2, \qquad \text{MAE} = \sum_{i=1}^{L}|\mathbf{X}_i - \hat{\mathbf{X}}_i|. \tag{17}$$

where $\mathbf{X}, \hat{\mathbf{X}} \in \mathbb{R}^{L \times N}$ denote the ground truth and prediction results for $N$ variates in the future $L$ time steps. $|\cdot|$ means the absolute value operation.

**Algorithm details**  We provide the pseudo-code of FADformer in Algorithm 1.

## D  ERROR BARS

We calculate the standard deviation of FADformer performance by training the model with five different random seeds across 12 datasets. As shown in Table 8, the error bars for all the results are small, indicating that our FADformer is robust and reliable.

---

[1] All the datasets are publicly available at `https://github.com/thuml/iTransformer`

---

**Algorithm 1** Workflow of our FADformer.

---

**Input:** Input lookback time series $\mathbf{X} \in \mathbb{R}^{T \times N}$; Input length $T$, prediction length $H$, and variates number $N$; Token dimension $D$, the number of FADformer blocks $L$, and top K largest magnitude frequencies $K$.

**Output:** The prediction results $\hat{\mathbf{X}} \in \mathbb{R}^{H \times N}$.

1: $\mathbf{X} = \mathbf{X}.transpose(-1, -2)$    $\triangleright \mathbf{X} \in \mathbb{R}^{T \times N}$
2: $\triangleright$ Embedding series into variate tokens by Multi-layer Perceptron.
3: $\mathbf{F}_0 = MLP(\mathbf{X})$    $\triangleright \mathbf{F}_0 \in \mathbb{R}^{N \times D}$
4: $\triangleright$ Running through FADformer blocks.
5: **for** $l$ in $\{1, \cdots, L\}$ **do**
6:    $\triangleright$ Debiasing modules are applied to the self-attention mechanism and residual connection.
7:    $\mathbf{Q}_{l-1} = Linear_1(\mathbf{F}_{l-1}), \mathbf{K}_{l-1} = Linear_2(\mathbf{F}_{l-1}), \mathbf{V}_{l-1} = Linear_3(\mathbf{F}_{l-1})$    $\triangleright$
   $\mathbf{Q}_{l-1}, \mathbf{K}_{l-1}, \mathbf{V}_{l-1} \in \mathbb{R}^{N \times h \times d}$
8:    $\triangleright$ $h$ is the number of heads, $d$ is the dimension for each head.
9:    $\mathbf{A}_{l-1} = softmax(\mathbf{Q}_{l-1}@(\mathbf{K}_{l-1}.transpose(-2, -1)))$    $\triangleright \mathbf{A}_{l-1} \in \mathbb{R}^{h \times N \times N}$
10:    $\triangleright$ Decomposing the low-pass component from $\mathbf{A}_{l-1}$.
11:    $\mathbf{A}_{l-1}^D = torch.ones(\mathbf{A}_{l-1}.shape[-2:])/N$    $\triangleright \mathbf{A}_{l-1}^D \in \mathbb{R}^{N \times N}$
12:    $\mathbf{A}_{l-1}^D = \mathbf{A}_{l-1}^D[None, \cdots]$    $\triangleright \mathbf{A}_{l-1}^D \in \mathbb{R}^{h \times N \times N}$
13:    $\mathbf{A}_{l-1}^H = \mathbf{A}_{l-1} - \mathbf{A}_{l-1}^D$    $\triangleright \mathbf{A}_{l-1}^H \in \mathbb{R}^{h \times N \times N}$
14:    $\triangleright$ Defining a learnable parameter $\lambda \in \mathbb{R}^h$ and rescaling the high-pass component of attention weights.
15:    $\mathbf{A}_{l-1}^H = \mathbf{A}_{l-1}^H * (1 + \lambda.reshape[:, None, None])$
16:    $\mathbf{A}'_{l-1} = \mathbf{A}_{l-1}^D + \mathbf{A}_{l-1}^H$    $\triangleright \mathbf{A}'_{l-1} \in \mathbb{R}^{h \times N \times N}$
17:    $\mathbf{F}_1 = \mathbf{A}'_{l-1}@\mathbf{V}_{l-1}.transpose(1, 2).reshape(N, -1)$    $\triangleright \mathbf{F}_1 \in \mathbb{R}^{N \times D}$
18:    $\triangleright$ Applying Fourier transform to each series to extract low- and high-components for embedded features.
19:    $\mathbf{Z} = torch.fft.rfft(\mathbf{F}_{l-1}, dim=1)$
20:    $k_s = torch.topk(\mathbf{Z}.abs(), K, dim=1), topk\_index = k_s.indices$
21:    $m = torch.zeros\_like(\mathbf{Z}), m.scatter\_(1, topk\_index, 1)$
22:    $\mathbf{Z}_m = \mathbf{Z} * m$
23:    $\mathbf{F}_d = torch.fft.irfft(\mathbf{Z}_m, dim-1).real(), \mathbf{F}_h = \mathbf{F}_{l-1} - \mathbf{F}_d$    $\triangleright \mathbf{F}_d, \mathbf{F}_h \in \mathbb{R}^{N \times D}$
24:    $\triangleright$ Defining two learnable parameter $\alpha, \beta \in \mathbb{R}^D$ and rescaling the weights for decomposed embedded features.
25:    $\mathbf{F}_2 = \mathbf{F}_{l-1} + \mathbf{F}_d * \alpha + \mathbf{F}_h * \beta$    $\triangleright \mathbf{F}_2 \in \mathbb{R}^{N \times D}$
26:    $\mathbf{F}_{l-1} = LayerNorm(\mathbf{F}_1 + \mathbf{F}_2)$    $\triangleright \mathbf{F}_{l-1} \in \mathbb{R}^{N \times D}$
27:    $\mathbf{F}_l = LayerNorm(\mathbf{F}_{l-1} + FeedForward(\mathbf{F}_{l-1}))$    $\triangleright \mathbf{F}_l \in \mathbb{R}^{N \times D}$
28: **end for**
29: $\hat{\mathbf{X}} = Projector(\mathbf{F}_l)$    $\triangleright \hat{\mathbf{X}} \in \mathbb{R}^{N \times H}$
30: $\hat{\mathbf{X}} = \hat{\mathbf{X}}.transpose(-1, -2)$    $\triangleright \hat{\mathbf{X}} \in \mathbb{R}^{H \times N}$

---

Table 7: Robustness of FADformer performance obtained from 5 random seeds on 12 benchmarks.

| Dataset | ETTm1 | | ETTm2 | | ETTh2 | | ECL | |
|---|---|---|---|---|---|---|---|---|
| Metrics | MSE | MAE | MSE | MAE | MSE | MAE | MSE | MAE |
| 96 | $0.319 \pm 0.001$ | $0.349 \pm 0.001$ | $0.174 \pm 0.000$ | $0.252 \pm 0.001$ | $0.293 \pm 0.001$ | $0.340 \pm 0.000$ | $0.141 \pm 0.000$ | $0.229 \pm 0.000$ |
| 192 | $0.370 \pm 0.001$ | $0.379 \pm 0.001$ | $0.239 \pm 0.001$ | $0.295 \pm 0.000$ | $0.368 \pm 0.002$ | $0.388 \pm 0.001$ | $0.156 \pm 0.001$ | $0.246 \pm 0.001$ |
| 336 | $0.403 \pm 0.001$ | $0.396 \pm 0.002$ | $0.303 \pm 0.001$ | $0.335 \pm 0.001$ | $0.419 \pm 0.004$ | $0.427 \pm 0.002$ | $0.171 \pm 0.002$ | $0.260 \pm 0.001$ |
| 720 | $0.469 \pm 0.002$ | $0.434 \pm 0.001$ | $0.399 \pm 0.002$ | $0.394 \pm 0.001$ | $0.422 \pm 0.003$ | $0.440 \pm 0.001$ | $0.194 \pm 0.002$ | $0.283 \pm 0.002$ |

| Dataset | Traffic | | Exchange | | Solar-Energy | | Weather | |
|---|---|---|---|---|---|---|---|---|
| Metrics | MSE | MAE | MSE | MAE | MSE | MAE | MSE | MAE |
| 96 | $0.386 \pm 0.001$ | $0.254 \pm 0.000$ | $0.084 \pm 0.001$ | $0.203 \pm 0.001$ | $0.199 \pm 0.002$ | $0.208 \pm 0.002$ | $0.162 \pm 0.000$ | $0.196 \pm 0.000$ |
| 192 | $0.403 \pm 0.002$ | $0.261 \pm 0.001$ | $0.176 \pm 0.002$ | $0.298 \pm 0.001$ | $0.223 \pm 0.001$ | $0.228 \pm 0.001$ | $0.212 \pm 0.001$ | $0.244 \pm 0.001$ |
| 336 | $0.414 \pm 0.003$ | $0.267 \pm 0.003$ | $0.329 \pm 0.002$ | $0.415 \pm 0.001$ | $0.245 \pm 0.000$ | $0.243 \pm 0.001$ | $0.272 \pm 0.002$ | $0.287 \pm 0.001$ |
| 720 | $0.445 \pm 0.001$ | $0.283 \pm 0.002$ | $0.827 \pm 0.012$ | $0.688 \pm 0.003$ | $0.247 \pm 0.001$ | $0.242 \pm 0.002$ | $0.351 \pm 0.002$ | $0.342 \pm 0.002$ |

| Dataset | PEMS03 | | PEMS04 | | PEMS07 | | PEMS08 | |
|---|---|---|---|---|---|---|---|---|
| Metrics | MSE | MAE | MSE | MAE | MSE | MAE | MSE | MAE |
| 12 | $0.067 \pm 0.000$ | $0.168 \pm 0.000$ | $0.075 \pm 0.001$ | $0.180 \pm 0.001$ | $0.070 \pm 0.000$ | $0.161 \pm 0.000$ | $0.077 \pm 0.000$ | $0.176 \pm 0.001$ |
| 24 | $0.087 \pm 0.000$ | $0.192 \pm 0.000$ | $0.091 \pm 0.001$ | $0.200 \pm 0.000$ | $0.088 \pm 0.000$ | $0.182 \pm 0.000$ | $0.110 \pm 0.000$ | $0.209 \pm 0.000$ |
| 48 | $0.122 \pm 0.001$ | $0.231 \pm 0.001$ | $0.118 \pm 0.002$ | $0.227 \pm 0.001$ | $0.111 \pm 0.001$ | $0.205 \pm 0.001$ | $0.174 \pm 0.001$ | $0.257 \pm 0.001$ |
| 96 | $0.155 \pm 0.002$ | $0.264 \pm 0.001$ | $0.149 \pm 0.001$ | $0.258 \pm 0.001$ | $0.138 \pm 0.001$ | $0.232 \pm 0.001$ | $0.226 \pm 0.002$ | $0.262 \pm 0.001$ |

Table 8: Generalization performance by applying various efficient attention mechanisms to our debiasing framework with the input length $T = 96$. For the prediction horizon, we set $H = 96$ for PEMS and $H = 720$ for the others, respectively.

| Attention Types | | Self-attention (2017) | | Prob-attention (2021) | | Auto-attention (2021) | | Flow-attention (2022) | | Period-attention (2023) | |
|---|---|---|---|---|---|---|---|---|---|---|---|
| Metrics | | MSE | MAE | MSE | MAE | MSE | MAE | MSE | MAE | MSE | MAE |
| ECL | Original | 0.302 | 0.386 | 0.373 | 0.439 | 0.254 | 0.361 | 0.296 | 0.380 | 0.293 | 0.390 |
| | +Debiasing | **0.196** | **0.283** | **0.205** | **0.303** | **0.218** | **0.301** | **0.202** | **0.292** | **0.210** | **0.299** |
| | **Improvement** | **35.1%** | **26.7%** | **45.0%** | **31.0%** | **14.2%** | **16.6%** | **31.8%** | **23.2%** | **28.3%** | **23.3%** |
| Traffic | Original | 0.697 | 0.376 | 0.864 | 0.472 | 0.660 | 0.408 | 0.825 | 0.449 | 0.672 | 0.423 |
| | +Debiasing | **0.445** | **0.283** | **0.469** | **0.305** | **0.465** | **0.302** | **0.472** | **0.303** | **0.462** | **0.308** |
| | **Improvement** | **36.2%** | **24.7%** | **45.7%** | **35.4%** | **29.5%** | **26.0%** | **42.8%** | **32.5%** | **31.3%** | **27.2%** |
| PEMS03 | Original | 0.175 | 0.274 | 0.211 | 0.304 | 1.031 | 0.796 | 0.173 | 0.287 | 0.270 | 0.385 |
| | +Debiasing | **0.155** | **0.264** | **0.165** | **0.276** | **0.367** | **0.369** | **0.164** | **0.273** | **0.155** | **0.264** |
| | **Improvement** | **11.4%** | **3.6%** | **21.8%** | **9.2%** | **64.4%** | **53.6%** | **5.2%** | **4.9%** | **44.9%** | **26.5%** |
| PEMS07 | Original | 0.185 | 0.252 | 0.196 | 0.260 | 0.554 | 0.578 | 0.180 | 0.245 | 0.309 | 0.407 |
| | +Debiasing | **0.138** | **0.232** | **0.143** | **0.238** | **0.335** | **0.362** | **0.140** | **0.238** | **0.202** | **0.293** |
| | **Improvement** | **25.7%** | **7.9%** | **27.0%** | **8.5%** | **39.5%** | **37.4%** | **22.2%** | **2.9%** | **34.6%** | **28.0%** |

# E EXTRA GENERALIZATION PERFORMANCE

To explore the generalization of FADformer under different attention mechanisms, we employ self-attention (Vaswani et al., 2017), prob-attention (Zhou et al., 2021), auto-attention (Wu et al., 2021), flow-attention (Wu et al., 2022), and period-attention (Liang et al., 2023) into our framework. The generalization performance for different attention mechanisms is recorded in Table 3, where all the attention mechanisms are adopted on the variate dimension for comparison. Overall, our debiasing modules achieve an average improvement of 21.4% on Self-attention, 28.0% on Prob-attention, 35.2% on Auto-attention, 20.7% on Flow-attention, and 30.5% on Period-attention for the ECL, Traffic, PEMS03, and PEMS07 benchmarks. Besides, we provide the full results of generalization on three SOTA Transformer-based forecasters in Table 9.

Table 9: Generalization performance on three SOTA Transformer-based models.

| Metric | | TimeBridge | | +Debiasing | | TQNet | | +Debiasing | | Leddam | | +Debiasing | |
|---|---|---|---|---|---|---|---|---|---|---|---|---|---|
| | | MSE | MAE | MSE | MAE | MSE | MAE | MSE | MAE | MSE | MAE | MSE | MAE |
| Weather | 96 | 0.144 | 0.184 | 0.142 | 0.183 | 0.157 | 0.200 | 0.155 | 0.199 | 0.156 | 0.202 | 0.152 | 0.198 |
| | 192 | 0.186 | 0.225 | 0.183 | 0.224 | 0.206 | 0.245 | 0.203 | 0.243 | 0.207 | 0.250 | 0.203 | 0.246 |
| | 336 | 0.237 | 0.267 | 0.234 | 0.265 | 0.262 | 0.287 | 0.259 | 0.285 | 0.262 | 0.291 | 0.258 | 0.287 |
| | 720 | 0.307 | 0.320 | 0.303 | 0.318 | 0.344 | 0.342 | 0.339 | 0.340 | 0.343 | 0.343 | 0.339 | 0.337 |
| | Avg | 0.219 | 0.249 | 0.216 | 0.248 | 0.242 | 0.269 | 0.239 | 0.267 | 0.242 | 0.272 | 0.238 | 0.267 |
| Solar | 96 | 0.161 | 0.224 | 0.159 | 0.223 | 0.173 | 0.233 | 0.172 | 0.230 | 0.197 | 0.241 | 0.196 | 0.239 |
| | 192 | 0.177 | 0.237 | 0.174 | 0.235 | 0.199 | 0.257 | 0.197 | 0.256 | 0.231 | 0.264 | 0.228 | 0.262 |
| | 336 | 0.188 | 0.244 | 0.185 | 0.242 | 0.211 | 0.263 | 0.208 | 0.261 | 0.241 | 0.268 | 0.239 | 0.265 |
| | 720 | 0.197 | 0.252 | 0.196 | 0.250 | 0.209 | 0.270 | 0.206 | 0.269 | 0.250 | 0.281 | 0.248 | 0.278 |
| | Avg | 0.181 | 0.239 | 0.179 | 0.238 | 0.198 | 0.256 | 0.196 | 0.254 | 0.230 | 0.264 | 0.228 | 0.261 |
| ECL | 96 | 0.120 | 0.214 | 0.118 | 0.213 | 0.134 | 0.229 | 0.132 | 0.227 | 0.141 | 0.235 | 0.138 | 0.232 |
| | 192 | 0.142 | 0.237 | 0.140 | 0.236 | 0.154 | 0.247 | 0.152 | 0.245 | 0.159 | 0.252 | 0.156 | 0.249 |
| | 336 | 0.156 | 0.252 | 0.155 | 0.250 | 0.169 | 0.264 | 0.165 | 0.263 | 0.173 | 0.268 | 0.169 | 0.263 |
| | 720 | 0.179 | 0.278 | 0.176 | 0.277 | 0.201 | 0.294 | 0.199 | 0.292 | 0.201 | 0.295 | 0.199 | 0.289 |
| | Avg | 0.149 | 0.245 | 0.147 | 0.244 | 0.164 | 0.259 | 0.162 | 0.257 | 0.169 | 0.263 | 0.166 | 0.258 |
| Traffic | 96 | 0.340 | 0.240 | 0.338 | 0.239 | 0.413 | 0.261 | 0.403 | 0.257 | 0.426 | 0.276 | 0.413 | 0.272 |
| | 192 | 0.343 | 0.250 | 0.341 | 0.249 | 0.432 | 0.271 | 0.418 | 0.265 | 0.458 | 0.289 | 0.432 | 0.283 |
| | 336 | 0.363 | 0.257 | 0.360 | 0.255 | 0.450 | 0.277 | 0.439 | 0.274 | 0.486 | 0.297 | 0.445 | 0.292 |
| | 720 | 0.393 | 0.271 | 0.389 | 0.269 | 0.486 | 0.295 | 0.468 | 0.289 | 0.498 | 0.313 | 0.462 | 0.305 |
| | Avg | 0.360 | 0.255 | 0.357 | 0.253 | 0.445 | 0.276 | 0.432 | 0.271 | 0.467 | 0.294 | 0.438 | 0.288 |

# F  Extra Ablation Studies

To further elaborate on the effectiveness of our FADformer, we provide more detailed results and analysis here for ablation studies.

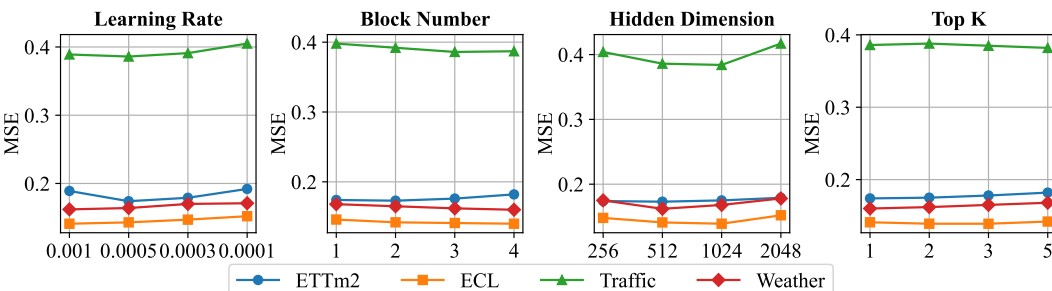

Figure 6: Hyperparameter sensitivity concerning the learning rate, the number of Transformer blocks, the hidden dimension of variate tokens, and the selection of $K$ for the Top K largest amplitude as low-frequency components. The results are recorded with an input length of $T = 96$ and a prediction length of $H = 96$ on four benchmarks.

## F.1  Hyperparameter Sensitivity

As seen in Figure 6, we evaluate the hyperparameter sensitivity of FADformer in terms of the learning rate, the number of Transformer blocks, the hidden dimension of variate tokens, and the selection of $K$ for the Top K largest amplitude as low-frequency components. Note that, since the selection of $K$ is related to the number of variates, we only take $K = \{1, 2, 3, 5\}$ to satisfy different datasets for analysis. The results fluctuate under different hyperparameter settings, indicating that the choice of hyperparameters is critical to the model. As the most common hyperparameter, the learning rate should be carefully selected for different datasets. In most cases, increasing the number of Transformer blocks tends to strengthen the model performance, especially for datasets with numerous variates. Moreover, larger hidden dimensions and more Transformer blocks are more effective for handling the intricacies of complex datasets. The selection of $K$ causes a very slight fluctuation in forecasting performance, which reveals that it is reasonable to regard the largest $K$ amplitudes obtained from the frequency domain as the low-frequency component of the features.

## F.2  Full ablation studies for debiasing modules

The comparison of different debiasing variants in our FADformer is shown in Table 10. For the AttnDeb module, **Mean** means that we compute the average value of attention weights as low-pass components, **Uniform** refers to capturing the ow-pass component of attention weights with uniform distribution, and **Gaussian** learns the low-pass weights by a Gaussian distribution. For the FeatDeb module, **Mean** indicates that the low-frequency components are obtained by the average value of all the series representations, **First-K** defines the first $K$ lowest elements of the Fourier transform as low-frequency components of features, and **Top-K** selects the frequency set with the top $K$ largest amplitude as low-frequency components. Compared to the performance of other Transformer-based baselines in Table 12, the results in Table 10 indicate that all the debiasing variants consistently enhance the model's performance. Additionally, our Gaussian and Top-K strategies are the optimal selections for our AttnDeb and FeatDeb, respectively.

## F.3  Performance Comparison with Anti-oversmoothing.

Anti-oversmoothing (Wang et al., 2022) is an excellent work that uses the AttnScale and FeatScale techniques to mitigate the undesirable low-pass limitation of the self-attention mechanism in the computer vision domain. To highlight the superiority of our method, we employ the AttnScale and FeatScale techniques in the time series forecasting task and compare their performance with FADformer.

Table 10: Full ablation results for different debiasing variants across four datasets.

| Variants | | | | Weather | | Solar | | ECL | | Traffic | |
|---|---|---|---|---|---|---|---|---|---|---|---|
| | | Metric | | MSE | MAE | MSE | MAE | MSE | MAE | MSE | MAE |
| AttnDeb | **Mean** | | 96 | 0.172 | 0.210 | 0.201 | 0.215 | 0.143 | 0.232 | 0.395 | 0.262 |
| | | | 192 | 0.220 | 0.241 | 0.235 | 0.230 | 0.160 | 0.247 | 0.412 | 0.273 |
| | | | 336 | 0.274 | 0.289 | 0.247 | 0.257 | 0.171 | 0.260 | 0.425 | 0.280 |
| | | | 720 | 0.355 | 0.347 | 0.249 | 0.264 | 0.207 | 0.284 | 0.446 | 0.292 |
| | | | Avg | 0.255 | 0.272 | 0.233 | 0.242 | 0.170 | 0.256 | 0.420 | 0.277 |
| | **Uniform** | | 96 | 0.167 | 0.200 | 0.205 | 0.211 | 0.142 | 0.235 | 0.403 | 0.261 |
| | | | 192 | 0.219 | 0.248 | 0.231 | 0.231 | 0.161 | 0.251 | 0.424 | 0.267 |
| | | | 336 | 0.278 | 0.292 | 0.248 | 0.245 | 0.172 | 0.263 | 0.428 | 0.274 |
| | | | 720 | 0.351 | 0.341 | 0.250 | 0.250 | 0.205 | 0.281 | 0.448 | 0.293 |
| | | | Avg | 0.254 | 0.270 | 0.234 | 0.234 | 0.170 | 0.258 | 0.426 | 0.274 |
| | **Gaussian** | | 96 | 0.162 | 0.196 | 0.199 | 0.208 | 0.141 | 0.229 | 0.386 | 0.275 |
| | | | 192 | 0.212 | 0.244 | 0.223 | 0.228 | 0.156 | 0.246 | 0.403 | 0.261 |
| | | | 336 | 0.272 | 0.287 | 0.245 | 0.243 | 0.171 | 0.260 | 0.414 | 0.267 |
| | | | 720 | 0.351 | 0.342 | 0.247 | 0.242 | 0.196 | 0.283 | 0.445 | 0.283 |
| | | | Avg | 0.249 | 0.267 | 0.229 | 0.230 | 0.166 | 0.255 | 0.412 | 0.266 |
| FeatDeb | **Mean** | | 96 | 0.168 | 0.207 | 0.204 | 0.212 | 0.145 | 0.235 | 0.391 | 0.259 |
| | | | 192 | 0.217 | 0.250 | 0.233 | 0.238 | 0.161 | 0.252 | 0.410 | 0.272 |
| | | | 336 | 0.275 | 0.291 | 0.246 | 0.252 | 0.172 | 0.266 | 0.424 | 0.279 |
| | | | 720 | 0.355 | 0.343 | 0.250 | 0.253 | 0.205 | 0.297 | 0.452 | 0.295 |
| | | | Avg | 0.254 | 0.273 | 0.233 | 0.239 | 0.171 | 0.263 | 0.419 | 0.276 |
| | **First-K** | | 96 | 0.170 | 0.210 | 0.210 | 0.215 | 0.143 | 0.233 | 0.392 | 0.258 |
| | | | 192 | 0.215 | 0.248 | 0.221 | 0.253 | 0.160 | 0.254 | 0.408 | 0.265 |
| | | | 336 | 0.278 | 0.296 | 0.254 | 0.249 | 0.175 | 0.262 | 0.419 | 0.277 |
| | | | 720 | 0.356 | 0.351 | 0.253 | 0.250 | 0.201 | 0.287 | 0.448 | 0.290 |
| | | | Avg | 0.255 | 0.276 | 0.235 | 0.242 | 0.170 | 0.259 | 0.417 | 0.273 |
| | **Top-K** | | 96 | 0.162 | 0.196 | 0.199 | 0.208 | 0.141 | 0.229 | 0.386 | 0.254 |
| | | | 192 | 0.212 | 0.244 | 0.223 | 0.228 | 0.156 | 0.246 | 0.403 | 0.261 |
| | | | 336 | 0.272 | 0.287 | 0.245 | 0.243 | 0.171 | 0.260 | 0.414 | 0.267 |
| | | | 720 | 0.351 | 0.342 | 0.247 | 0.242 | 0.196 | 0.283 | 0.445 | 0.283 |
| | | | Avg | 0.249 | 0.267 | 0.229 | 0.230 | 0.166 | 0.255 | 0.412 | 0.266 |

Specifically, we utilize iTransformer (Liu et al., 2024) as the backbone and employ anti-oversmoothing techniques (i.e., AttnScale and FeatScale) along with our debiasing strategies (i.e., AttnDeb and FeatDeb) to evaluate the forecasting performance.

Table 11: Forecasting performance comparison with Anti-oversmoothing (Wang et al., 2022).

| Metric | | iTransformer | | +AttnScale | | +FeatScale | | +AttnDeb | | +FeatDeb | | +AttnDeb & FeatDeb | |
|---|---|---|---|---|---|---|---|---|---|---|---|---|---|
| | | MSE | MAE | MSE | MAE | MSE | MAE | MSE | MAE | MSE | MAE | MSE | MAE |
| ECL | 96 | 0.148 | 0.240 | 0.152 | 0.246 | 0.153 | 0.248 | 0.145 | 0.236 | 0.143 | 0.235 | 0.141 | 0.229 |
| | 192 | 0.162 | 0.253 | 0.165 | 0.258 | 0.163 | 0.255 | 0.160 | 0.250 | 0.159 | 0.249 | 0.156 | 0.246 |
| | 336 | 0.178 | 0.269 | 0.179 | 0.272 | 0.180 | 0.276 | 0.175 | 0.264 | 0.176 | 0.265 | 0.171 | 0.260 |
| | 720 | 0.225 | 0.317 | 0.230 | 0.323 | 0.228 | 0.320 | 0.210 | 0.296 | 0.206 | 0.293 | 0.196 | 0.283 |
| | Avg | 0.178 | 0.270 | 0.182 | 0.275 | 0.181 | 0.275 | 0.173 | 0.262 | 0.171 | 0.261 | 0.166 | 0.255 |
| Traffic | 96 | 0.395 | 0.268 | 0.394 | 0.267 | 0.395 | 0.269 | 0.390 | 0.263 | 0.389 | 0.259 | 0.386 | 0.254 |
| | 192 | 0.417 | 0.276 | 0.415 | 0.274 | 0.416 | 0.275 | 0.410 | 0.272 | 0.408 | 0.270 | 0.403 | 0.261 |
| | 336 | 0.433 | 0.283 | 0.429 | 0.279 | 0.430 | 0.280 | 0.420 | 0.274 | 0.421 | 0.275 | 0.414 | 0.267 |
| | 720 | 0.467 | 0.302 | 0.460 | 0.299 | 0.459 | 0.298 | 0.456 | 0.290 | 0.458 | 0.292 | 0.445 | 0.283 |
| | Avg | 0.428 | 0.282 | 0.425 | 0.280 | 0.425 | 0.281 | 0.419 | 0.275 | 0.419 | 0.274 | 0.412 | 0.266 |

As shown in Table 11, the AttnScale and FeatScale methods from Anti-oversmoothing (Liu et al., 2024) demonstrate limited performance improvements in time series forecasting tasks, performing even worse than the backbone model, iTransformer. In contrast, our proposed debiasing strategies, AttnDeb and FeatDeb, significantly enhance prediction performance. Specifically, when used in combination, AttnDeb and FeatDeb achieve the best overall results.

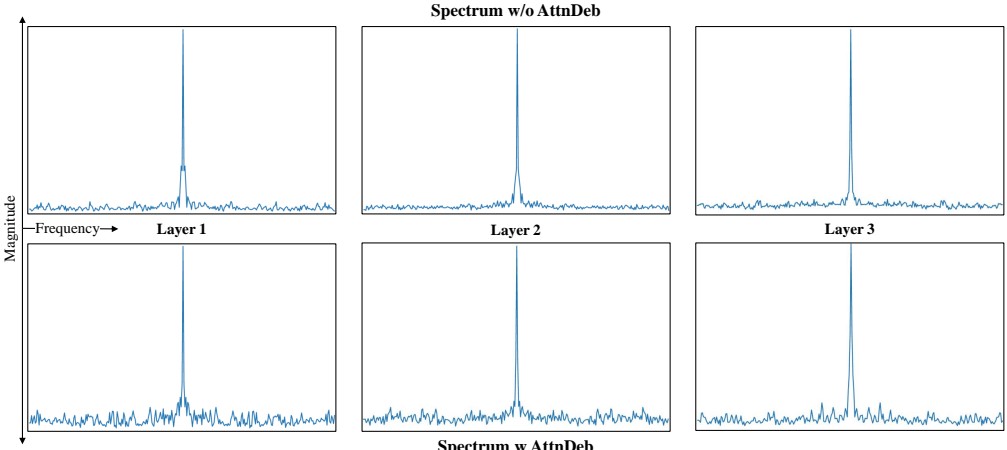

Figure 7: Visualize the spectrum of the attention matrix w or w/o AttnDeb on ECL. We randomly sample the spectral response of the attention matrix at the first head for a clear comparison.

## G SHOWCASES

### G.1 VISUALIZATION ON SPECTRUM AND CORRELATION

We provide additional visualization of the attention matrix spectrum to further validate our debiasing effect for the AttnDeb module. We compute the spectrum of the attention map $\mathbf{A}$ by regarding $\mathbf{A}$ as a linear filter, where its Fourier domain response is another linear kernel $\mathbf{\Lambda} = \mathcal{F}\mathbf{A}\mathcal{F}^{-1}$. For a spectrum $\tilde{X} = \mathcal{F}X$ of signals $X$, the $i$-th frequency response will be $\mathbf{\Lambda_i}\tilde{\mathbf{X}}$, where $\mathbf{\Lambda_i}$ is the $i$-th row of $\mathbf{\Lambda}$. Therefore, we can use $\|\mathbf{\Lambda_i}\|_{\mathbf{2}}$ to evaluate intensity of the $i$-th frequency band. In Figure 7, we plot the spectrum of 3-layer attention maps with/without our AttnDeb. We can find that the spectrum of attention maps reflects an obvious low-pass filtering effect, which is consistent with our Theorem A.1. Moreover, the attention spectrum of AttnDeb enjoys a richer response in the high-frequency band, indicating that our AttnDeb successfully elevates the high-frequency components against the dominance of low-pass components in the attention mechanism.

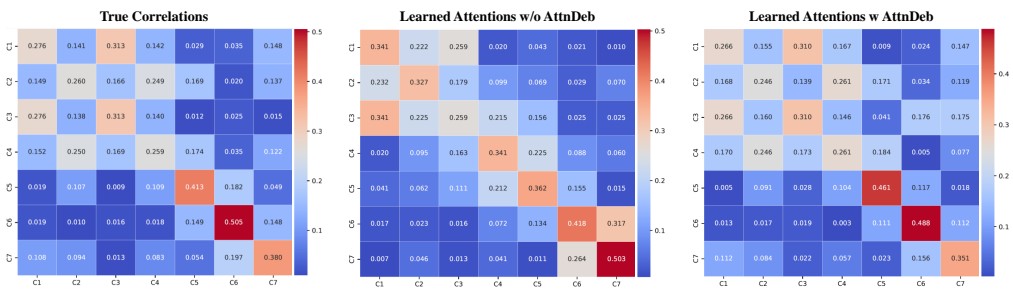

Figure 8: Visualize the weights of the attention matrix w or w/o AttnDeb on ECL. We randomly select seven channels of the attention matrix for a clear comparison. Please zoom in for more details.

To better illustrate the role of our AttnDeb, we compare the learned attention weights with the true correlation weights in Figure 8. We can observe that the true correlation weight matrix is not concentrated near the diagonal. For example, the weights in the bottom left and top right corners of the true correlation matrix are not negligible, both reflect the important dependencies between variate 1 and 7. However, the attention mechanism without our AttnDeb tends to focus on the large weights near the diagonal and indiscriminately filters out the dependencies far away from the diagonal. In contrast, our AttnDeb can capture useful information far away from the diagonal to compensate for the limitation of the attention mechanism. Therefore, the learned attention weights with our AttnDeb are more consistent with the true correlation distribution. We also visualize the

feature maps with/without FeatDeb at different layers for a clear comparison in Figure 9, which further reveals that our FeatDeb can capture more useful feature variations.

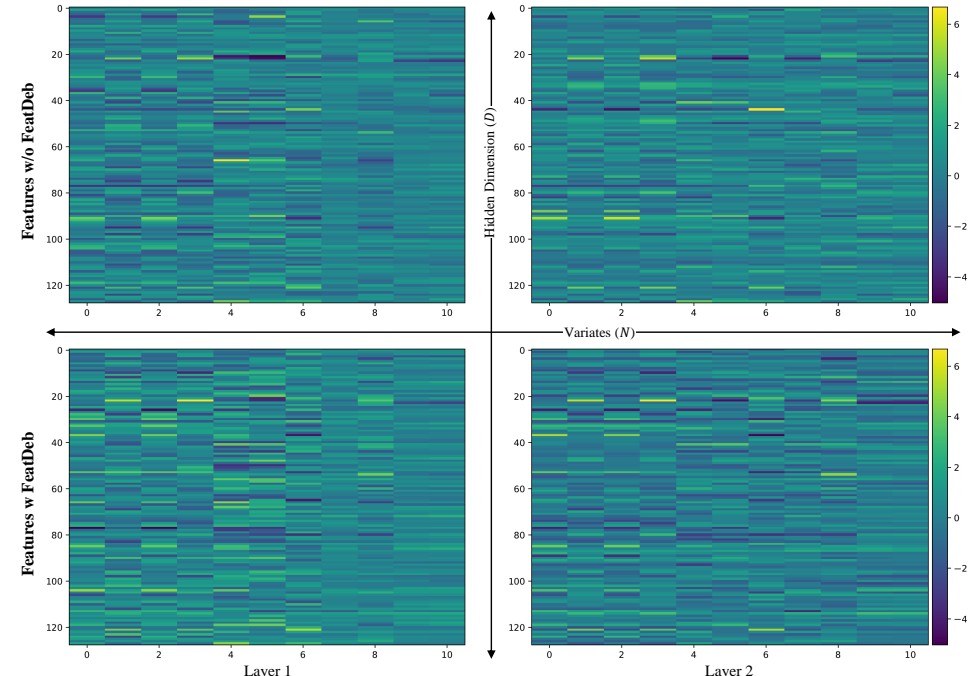

Figure 9: Visualize the feature maps w or w/o FeatDeb on ECL. We randomly sample the feature maps from ten variates with 128 hidden dimensions for a clear comparison.

## G.2 Visualization of Forecasting

To provide a more intuitive forecasting performance of FADformer, we present forecasting showcases on Weather (Figure 10), ECL (Figure 11), and Traffic (Figure 12) benchmarks compared with SOFTS (Han et al., 2024), iTransformer (Liu et al., 2024), PatchTST (Nie et al., 2023), Crossformer (Zhang & Yan, 2023), and DLinear (Zeng et al., 2023).

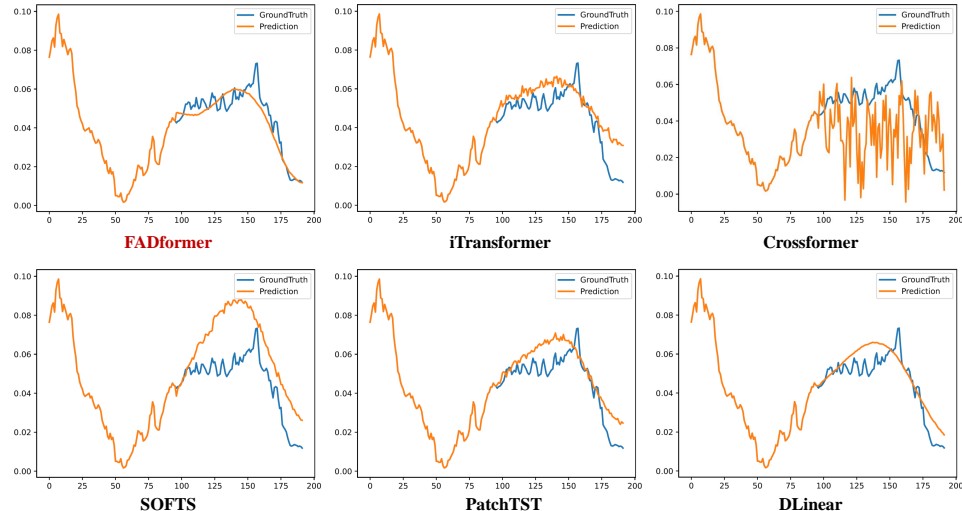

Figure 10: Visualization of input-96-predict-96 results on the Weather dataset.

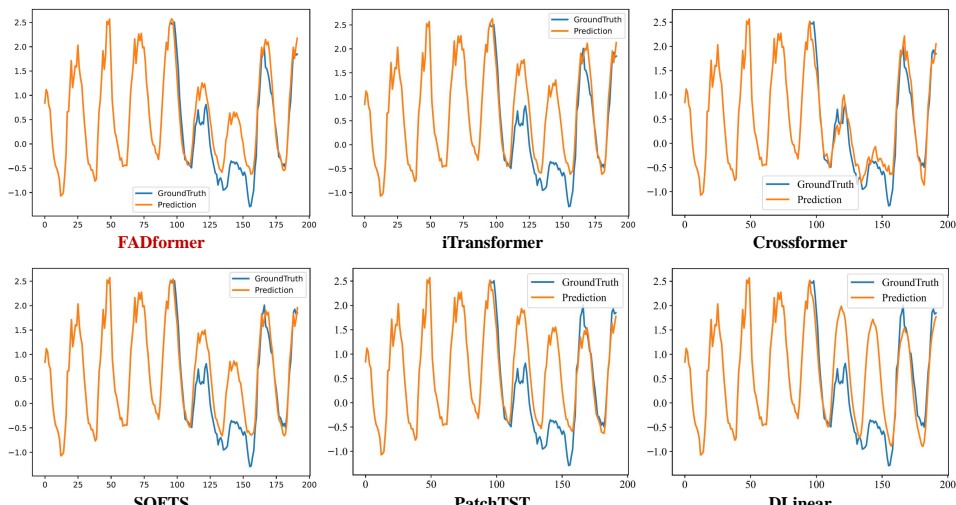

Figure 11: Visualization of input-96-predict-96 results on the ECL dataset.

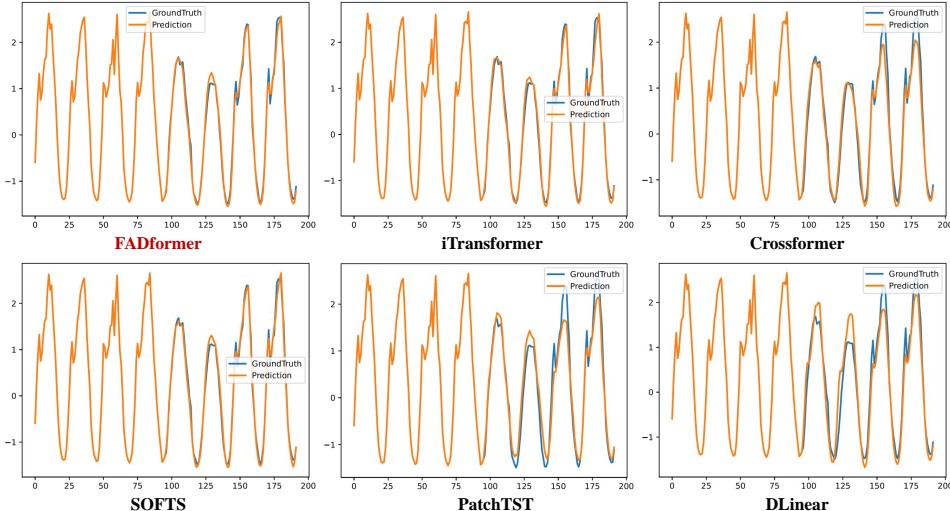

Figure 12: Visualization of input-96-predict-96 results on the Traffic dataset.

## H   FULL MAIN RESULTS

The complete results of our forecasting performance are presented in Table 12. We conducted experiments using 13 widely utilized real-world datasets and compared our method against eight previous state-of-the-art models.

Table 12: Full results of multivariate time series forecasting task.

| Models | | Metric | FADformer (Ours) | | SOFTS (2024) | | iTransformer (2024) | | PatchTST (2023) | | Crossformer (2023) | | DLinear (2023) | | TimesNet (2023) | | SCINet (2022) | | FEDformer (2022) | |
|---|---|---|---|---|---|---|---|---|---|---|---|---|---|---|---|---|---|---|---|---|
| | | | MSE | MAE | MSE | MAE | MSE | MAE | MSE | MAE | MSE | MAE | MSE | MAE | MSE | MAE | MSE | MAE | MSE | MAE |
| ETTh1 | 96 | | 0.378 | 0.394 | 0.382 | 0.399 | 0.386 | 0.405 | 0.394 | 0.406 | 0.423 | 0.448 | 0.386 | 0.400 | 0.384 | 0.402 | 0.654 | 0.599 | 0.376 | 0.419 |
| | 192 | | 0.431 | 0.422 | 0.435 | 0.431 | 0.441 | 0.436 | 0.440 | 0.435 | 0.471 | 0.474 | 0.437 | 0.432 | 0.436 | 0.429 | 0.719 | 0.631 | 0.420 | 0.448 |
| | 336 | | 0.475 | 0.445 | 0.480 | 0.452 | 0.487 | 0.458 | 0.491 | 0.462 | 0.570 | 0.546 | 0.481 | 0.459 | 0.491 | 0.469 | 0.778 | 0.659 | 0.459 | 0.465 |
| | 720 | | 0.489 | 0.475 | 0.499 | 0.488 | 0.503 | 0.491 | 0.487 | 0.479 | 0.653 | 0.621 | 0.519 | 0.516 | 0.521 | 0.500 | 0.836 | 0.699 | 0.506 | 0.507 |
| | Avg | | 0.443 | 0.434 | 0.449 | 0.443 | 0.454 | 0.448 | 0.453 | 0.446 | 0.529 | 0.522 | 0.456 | 0.452 | 0.458 | 0.450 | 0.747 | 0.647 | 0.440 | 0.460 |
| ETTh2 | 96 | | 0.293 | 0.340 | 0.297 | 0.347 | 0.297 | 0.349 | 0.288 | 0.340 | 0.745 | 0.584 | 0.333 | 0.387 | 0.340 | 0.374 | 0.707 | 0.621 | 0.358 | 0.397 |
| | 192 | | 0.368 | 0.388 | 0.373 | 0.394 | 0.380 | 0.400 | 0.376 | 0.395 | 0.877 | 0.656 | 0.477 | 0.476 | 0.402 | 0.414 | 0.860 | 0.689 | 0.429 | 0.439 |
| | 336 | | 0.419 | 0.427 | 0.415 | 0.426 | 0.428 | 0.432 | 0.440 | 0.451 | 1.043 | 0.731 | 0.594 | 0.541 | 0.452 | 0.452 | 1.000 | 0.744 | 0.496 | 0.487 |
| | 720 | | 0.422 | 0.440 | 0.421 | 0.433 | 0.427 | 0.445 | 0.436 | 0.453 | 1.104 | 0.763 | 0.831 | 0.657 | 0.462 | 0.468 | 1.249 | 0.838 | 0.463 | 0.474 |
| | Avg | | 0.376 | 0.399 | 0.377 | 0.400 | 0.383 | 0.407 | 0.385 | 0.410 | 0.942 | 0.684 | 0.599 | 0.515 | 0.414 | 0.427 | 0.954 | 0.723 | 0.437 | 0.449 |
| ETTm1 | 96 | | 0.319 | 0.349 | 0.325 | 0.361 | 0.334 | 0.368 | 0.329 | 0.365 | 0.404 | 0.426 | 0.345 | 0.372 | 0.338 | 0.375 | 0.418 | 0.438 | 0.379 | 0.419 |
| | 192 | | 0.370 | 0.379 | 0.370 | 0.385 | 0.377 | 0.391 | 0.380 | 0.394 | 0.450 | 0.451 | 0.380 | 0.389 | 0.374 | 0.387 | 0.439 | 0.450 | 0.426 | 0.441 |
| | 336 | | 0.403 | 0.396 | 0.412 | 0.412 | 0.426 | 0.420 | 0.400 | 0.410 | 0.532 | 0.515 | 0.413 | 0.413 | 0.410 | 0.411 | 0.490 | 0.485 | 0.445 | 0.459 |
| | 720 | | 0.469 | 0.434 | 0.472 | 0.448 | 0.491 | 0.459 | 0.475 | 0.453 | 0.666 | 0.589 | 0.474 | 0.453 | 0.478 | 0.450 | 0.595 | 0.550 | 0.543 | 0.490 |
| | Avg | | 0.390 | 0.390 | 0.395 | 0.402 | 0.407 | 0.410 | 0.396 | 0.406 | 0.513 | 0.495 | 0.403 | 0.407 | 0.400 | 0.406 | 0.486 | 0.481 | 0.448 | 0.452 |
| ETTm2 | 96 | | 0.174 | 0.252 | 0.180 | 0.261 | 0.180 | 0.264 | 0.184 | 0.264 | 0.287 | 0.366 | 0.193 | 0.292 | 0.187 | 0.267 | 0.286 | 0.377 | 0.203 | 0.287 |
| | 192 | | 0.239 | 0.295 | 0.246 | 0.306 | 0.250 | 0.309 | 0.246 | 0.306 | 0.414 | 0.492 | 0.284 | 0.362 | 0.249 | 0.309 | 0.399 | 0.445 | 0.269 | 0.328 |
| | 336 | | 0.303 | 0.335 | 0.316 | 0.350 | 0.311 | 0.348 | 0.308 | 0.346 | 0.597 | 0.542 | 0.369 | 0.427 | 0.321 | 0.351 | 0.637 | 0.591 | 0.325 | 0.366 |
| | 720 | | 0.399 | 0.394 | 0.410 | 0.403 | 0.412 | 0.407 | 0.409 | 0.402 | 1.730 | 1.042 | 0.554 | 0.522 | 0.408 | 0.403 | 0.960 | 0.735 | 0.421 | 0.415 |
| | Avg | | 0.279 | 0.319 | 0.288 | 0.330 | 0.288 | 0.332 | 0.287 | 0.330 | 0.757 | 0.611 | 0.350 | 0.401 | 0.291 | 0.333 | 0.571 | 0.537 | 0.305 | 0.349 |
| ECL | 96 | | 0.141 | 0.229 | 0.143 | 0.233 | 0.148 | 0.240 | 0.164 | 0.251 | 0.219 | 0.314 | 0.197 | 0.282 | 0.168 | 0.272 | 0.247 | 0.345 | 0.193 | 0.308 |
| | 192 | | 0.156 | 0.246 | 0.158 | 0.248 | 0.162 | 0.253 | 0.173 | 0.262 | 0.231 | 0.322 | 0.196 | 0.285 | 0.184 | 0.289 | 0.257 | 0.355 | 0.201 | 0.315 |
| | 336 | | 0.171 | 0.260 | 0.178 | 0.269 | 0.178 | 0.269 | 0.190 | 0.279 | 0.246 | 0.337 | 0.209 | 0.301 | 0.198 | 0.300 | 0.269 | 0.369 | 0.214 | 0.329 |
| | 720 | | 0.196 | 0.283 | 0.218 | 0.305 | 0.225 | 0.317 | 0.230 | 0.313 | 0.280 | 0.363 | 0.245 | 0.333 | 0.220 | 0.320 | 0.299 | 0.390 | 0.246 | 0.355 |
| | Avg | | 0.166 | 0.255 | 0.174 | 0.264 | 0.178 | 0.270 | 0.189 | 0.276 | 0.244 | 0.334 | 0.212 | 0.300 | 0.193 | 0.295 | 0.268 | 0.365 | 0.214 | 0.327 |
| Traffic | 96 | | 0.386 | 0.254 | 0.396 | 0.266 | 0.395 | 0.268 | 0.427 | 0.272 | 0.522 | 0.290 | 0.650 | 0.396 | 0.593 | 0.321 | 0.788 | 0.499 | 0.587 | 0.366 |
| | 192 | | 0.403 | 0.261 | 0.413 | 0.274 | 0.417 | 0.276 | 0.454 | 0.289 | 0.530 | 0.293 | 0.598 | 0.370 | 0.617 | 0.336 | 0.789 | 0.505 | 0.604 | 0.373 |
| | 336 | | 0.414 | 0.267 | 0.429 | 0.272 | 0.433 | 0.283 | 0.450 | 0.282 | 0.558 | 0.305 | 0.605 | 0.373 | 0.629 | 0.336 | 0.797 | 0.508 | 0.621 | 0.383 |
| | 720 | | 0.445 | 0.283 | 0.447 | 0.287 | 0.467 | 0.302 | 0.484 | 0.301 | 0.589 | 0.328 | 0.645 | 0.394 | 0.640 | 0.350 | 0.841 | 0.523 | 0.626 | 0.382 |
| | Avg | | 0.412 | 0.266 | 0.421 | 0.275 | 0.428 | 0.282 | 0.454 | 0.286 | 0.550 | 0.304 | 0.625 | 0.383 | 0.620 | 0.336 | 0.804 | 0.509 | 0.610 | 0.376 |
| Solar | 96 | | 0.199 | 0.208 | 0.200 | 0.232 | 0.203 | 0.237 | 0.205 | 0.246 | 0.310 | 0.331 | 0.290 | 0.378 | 0.250 | 0.292 | 0.237 | 0.344 | 0.242 | 0.342 |
| | 192 | | 0.223 | 0.228 | 0.229 | 0.254 | 0.233 | 0.261 | 0.237 | 0.267 | 0.734 | 0.725 | 0.320 | 0.398 | 0.296 | 0.318 | 0.280 | 0.380 | 0.285 | 0.380 |
| | 336 | | 0.245 | 0.243 | 0.242 | 0.268 | 0.248 | 0.273 | 0.250 | 0.276 | 0.750 | 0.735 | 0.353 | 0.415 | 0.319 | 0.330 | 0.304 | 0.389 | 0.282 | 0.376 |
| | 720 | | 0.247 | 0.242 | 0.248 | 0.271 | 0.249 | 0.275 | 0.252 | 0.275 | 0.769 | 0.765 | 0.356 | 0.413 | 0.338 | 0.337 | 0.308 | 0.388 | 0.357 | 0.427 |
| | Avg | | 0.229 | 0.230 | 0.230 | 0.256 | 0.233 | 0.262 | 0.236 | 0.266 | 0.641 | 0.639 | 0.330 | 0.401 | 0.301 | 0.319 | 0.282 | 0.375 | 0.261 | 0.381 |
| Weather | 96 | | 0.162 | 0.196 | 0.172 | 0.208 | 0.174 | 0.214 | 0.176 | 0.217 | 0.158 | 0.230 | 0.196 | 0.255 | 0.172 | 0.220 | 0.221 | 0.306 | 0.217 | 0.296 |
| | 192 | | 0.212 | 0.244 | 0.221 | 0.256 | 0.221 | 0.254 | 0.221 | 0.256 | 0.206 | 0.277 | 0.237 | 0.296 | 0.219 | 0.261 | 0.261 | 0.340 | 0.276 | 0.336 |
| | 336 | | 0.272 | 0.287 | 0.282 | 0.300 | 0.278 | 0.296 | 0.275 | 0.296 | 0.272 | 0.335 | 0.283 | 0.335 | 0.280 | 0.306 | 0.309 | 0.378 | 0.339 | 0.380 |
| | 720 | | 0.351 | 0.342 | 0.358 | 0.350 | 0.358 | 0.347 | 0.352 | 0.346 | 0.398 | 0.418 | 0.345 | 0.381 | 0.365 | 0.359 | 0.377 | 0.427 | 0.403 | 0.428 |
| | Avg | | 0.249 | 0.267 | 0.258 | 0.279 | 0.258 | 0.278 | 0.256 | 0.279 | 0.259 | 0.315 | 0.265 | 0.317 | 0.259 | 0.287 | 0.292 | 0.363 | 0.309 | 0.360 |
| Exchange | 96 | | 0.084 | 0.203 | 0.087 | 0.206 | 0.086 | 0.206 | 0.088 | 0.205 | 0.256 | 0.367 | 0.088 | 0.218 | 0.107 | 0.234 | 0.267 | 0.396 | 0.148 | 0.278 |
| | 192 | | 0.176 | 0.298 | 0.182 | 0.304 | 0.177 | 0.299 | 0.176 | 0.299 | 0.470 | 0.509 | 0.176 | 0.315 | 0.226 | 0.344 | 0.351 | 0.459 | 0.271 | 0.315 |
| | 336 | | 0.329 | 0.415 | 0.348 | 0.429 | 0.331 | 0.417 | 0.301 | 0.397 | 1.268 | 0.883 | 0.313 | 0.427 | 0.367 | 0.448 | 1.324 | 0.853 | 0.460 | 0.427 |
| | 720 | | 0.827 | 0.688 | 0.879 | 0.708 | 0.847 | 0.691 | 0.901 | 0.714 | 1.767 | 1.068 | 0.839 | 0.695 | 0.964 | 0.746 | 1.058 | 0.797 | 1.195 | 0.695 |
| | Avg | | 0.354 | 0.401 | 0.374 | 0.412 | 0.360 | 0.403 | 0.367 | 0.404 | 0.940 | 0.707 | 0.354 | 0.414 | 0.416 | 0.443 | 0.750 | 0.626 | 0.519 | 0.429 |
| PEMS03 | 12 | | 0.067 | 0.168 | 0.068 | 0.171 | 0.069 | 0.175 | 0.099 | 0.216 | 0.090 | 0.203 | 0.122 | 0.243 | 0.085 | 0.192 | 0.066 | 0.172 | 0.126 | 0.251 |
| | 24 | | 0.087 | 0.192 | 0.088 | 0.196 | 0.097 | 0.208 | 0.142 | 0.259 | 0.121 | 0.240 | 0.201 | 0.317 | 0.118 | 0.223 | 0.085 | 0.198 | 0.149 | 0.275 |
| | 48 | | 0.122 | 0.231 | 0.124 | 0.234 | 0.131 | 0.243 | 0.211 | 0.319 | 0.202 | 0.317 | 0.333 | 0.425 | 0.155 | 0.260 | 0.127 | 0.238 | 0.227 | 0.348 |
| | 96 | | 0.155 | 0.264 | 0.168 | 0.276 | 0.168 | 0.279 | 0.269 | 0.370 | 0.262 | 0.367 | 0.457 | 0.515 | 0.228 | 0.317 | 0.178 | 0.287 | 0.348 | 0.434 |
| | Avg | | 0.108 | 0.214 | 0.112 | 0.219 | 0.116 | 0.226 | 0.180 | 0.291 | 0.169 | 0.282 | 0.278 | 0.375 | 0.147 | 0.248 | 0.114 | 0.224 | 0.213 | 0.327 |
| PEMS04 | 12 | | 0.075 | 0.180 | 0.079 | 0.186 | 0.081 | 0.188 | 0.105 | 0.224 | 0.098 | 0.218 | 0.148 | 0.272 | 0.087 | 0.195 | 0.073 | 0.177 | 0.138 | 0.262 |
| | 24 | | 0.091 | 0.200 | 0.098 | 0.210 | 0.099 | 0.211 | 0.153 | 0.275 | 0.131 | 0.256 | 0.224 | 0.340 | 0.103 | 0.215 | 0.084 | 0.193 | 0.177 | 0.293 |
| | 48 | | 0.118 | 0.227 | 0.124 | 0.239 | 0.133 | 0.247 | 0.229 | 0.339 | 0.205 | 0.326 | 0.355 | 0.437 | 0.136 | 0.250 | 0.099 | 0.211 | 0.270 | 0.368 |
| | 96 | | 0.149 | 0.258 | 0.154 | 0.267 | 0.172 | 0.283 | 0.291 | 0.389 | 0.402 | 0.457 | 0.452 | 0.504 | 0.190 | 0.303 | 0.114 | 0.227 | 0.341 | 0.427 |
| | Avg | | 0.108 | 0.216 | 0.114 | 0.226 | 0.121 | 0.232 | 0.195 | 0.307 | 0.209 | 0.314 | 0.295 | 0.388 | 0.129 | 0.241 | 0.093 | 0.202 | 0.231 | 0.337 |
| PEMS07 | 12 | | 0.070 | 0.161 | 0.065 | 0.163 | 0.067 | 0.167 | 0.095 | 0.207 | 0.094 | 0.200 | 0.115 | 0.242 | 0.082 | 0.181 | 0.068 | 0.171 | 0.109 | 0.225 |
| | 24 | | 0.088 | 0.182 | 0.086 | 0.189 | 0.086 | 0.189 | 0.150 | 0.262 | 0.139 | 0.247 | 0.210 | 0.329 | 0.101 | 0.204 | 0.119 | 0.225 | 0.125 | 0.244 |
| | 48 | | 0.111 | 0.205 | 0.111 | 0.216 | 0.110 | 0.214 | 0.253 | 0.340 | 0.311 | 0.369 | 0.398 | 0.458 | 0.134 | 0.238 | 0.149 | 0.237 | 0.165 | 0.288 |
| | 96 | | 0.138 | 0.232 | 0.139 | 0.244 | 0.138 | 0.244 | 0.346 | 0.404 | 0.396 | 0.442 | 0.594 | 0.553 | 0.181 | 0.279 | 0.141 | 0.304 | 0.262 | 0.376 |
| | Avg | | 0.102 | 0.195 | 0.100 | 0.203 | 0.100 | 0.204 | 0.211 | 0.303 | 0.235 | 0.315 | 0.329 | 0.396 | 0.125 | 0.226 | 0.119 | 0.234 | 0.165 | 0.283 |
| PEMS08 | 12 | | 0.077 | 0.176 | 0.079 | 0.181 | 0.080 | 0.183 | 0.168 | 0.232 | 0.165 | 0.214 | 0.154 | 0.276 | 0.112 | 0.212 | 0.087 | 0.184 | 0.173 | 0.273 |
| | 24 | | 0.110 | 0.209 | 0.116 | 0.218 | 0.118 | 0.221 | 0.224 | 0.281 | 0.215 | 0.260 | 0.248 | 0.353 | 0.141 | 0.238 | 0.122 | 0.221 | 0.210 | 0.301 |
| | 48 | | 0.174 | 0.257 | 0.168 | 0.258 | 0.186 | 0.265 | 0.321 | 0.354 | 0.315 | 0.355 | 0.440 | 0.470 | 0.198 | 0.283 | 0.189 | 0.270 | 0.320 | 0.394 |
| | 96 | | 0.226 | 0.262 | 0.230 | 0.277 | 0.221 | 0.267 | 0.408 | 0.417 | 0.377 | 0.397 | 0.674 | 0.565 | 0.320 | 0.351 | 0.236 | 0.300 | 0.442 | 0.465 |
| | Avg | | 0.147 | 0.226 | 0.148 | 0.234 | 0.151 | 0.234 | 0.280 | 0.321 | 0.268 | 0.307 | 0.379 | 0.416 | 0.193 | 0.271 | 0.159 | 0.244 | 0.286 | 0.358 |
| 1st Count | | | 40 | 57 | 7 | 2 | 5 | 0 | 4 | 2 | 4 | 0 | 2 | 0 | 0 | 0 | 8 | 5 | 4 | 0 |

## I  MORE DISCUSSIONS AND EXTRA RESULTS

**Frequency-domain Modeling for Multivariate Time Series Forecasting.** Frequency-domain modeling is a powerful technique increasingly adopted for time-series forecasting. Specifically, Autoformer (Wu et al., 2021) replaces the self-attention with a Fast Fourier Transform (FFT)-based auto-correlation mechanism. FEDformer (Zhou et al., 2022) introduces frequency-enhanced attention to capture more detailed structures. FreTS (Yi et al., 2023) adopts an MLP-based frequency-domain framework to identify key features and patterns while handling complex and noisy data. FreDF (Wang et al., 2025) mitigates label autocorrelation by learning to forecast in the frequency domain. Nevertheless, the above methods tend to learn low-frequency features in the data and overlook high-frequency features, resulting in sub-optimal forecasting performance. To better capture high-frequency information, Fredformer (Piao et al., 2024) learns features equally across different frequency bands with a frequency-based transformer architecture, while Amplifier (Fei et al., 2025) brings attention to neglected low-energy components with an energy amplification technique. Unlike frequency debiasing and energy amplification, which are data-centric modeling strategies, we delve into the model itself and focus on the oversmoothing issue of Transformer-based models for time series forecasting. On the one hand, we analyze the low-pass filter effect of self-attention and rescale the attention weights via the AttnDeb to mitigate the low-pass limitation. On the other hand, we find out the rank collapse (Dong et al., 2021; Noci et al., 2022) of features and introduce the Effective Rank theory (Guo et al., 2023) and inject inductive feature bias into residual connections to amplify the important high-frequency features.

**Extra Results Compared with More Baselines.** To make a fair comparison with more baselines, we reproduced all the methods based on the official code repositories they provided under the same environment. During the training process, the input length is fixed to 96. Table 13 provides the full comparison results of FADformer against 8 related methods across 13 real-world datasets. The results show that FADformer consistently achieves state-of-the-art forecasting performance under most experimental settings, which underscores the effectiveness of the proposed method.

**Extra Results conducted on More Datasets.** We evaluate the performance of FADformer on the following real-world datasets from GIFT-Eval (Aksu et al., 2024):

- NASDAQ [2] comprises daily NASDAQ index data, combined with key economic indicators (e.g., interest rates, exchange rates, gold prices) from 2010 to 2024.

- Wiki [3] contains daily page views for 60,000 Wikipedia articles in eight different languages over two years (2018–2019). We select the first 99 articles as our experimental dataset.

- Covid-19 (Chen et al., 2022) includes daily COVID-19 hospitalization data in California (CA) from February to December 2020, provided by Johns Hopkins University.

- Wind (Li et al., 2022) contains hourly predicted wind speed data for a specific location, recorded at 15-minute intervals from January 1, 2020, to February 1, 2020.

- CzeLan (Qiu et al., 2024a) contains time-series monitoring data from the Czech Republic (CZE) collected between May and June 2016, which is recorded at 30-minute intervals.

- ZafNoo (Qiu et al., 2024a) contains solar irradiance measurements at consistent 30-minute intervals, spanning from mid-May to late June 2008.

To ensure fair comparisons, we choose a fixed data split ratio for each dataset chronologically, i.e., 7:1:2 or 6:2:2, for training, validation and testing. Due to the total length is insufficient in NASDAQ, Wiki, and Covid-19, we conduct forecasting results under the 'input-36-predict-24,36,48,60' setting. For other datasets, we set the input length and the forecasting lengths to 96 and $\{96, 192, 336, 720\}$, respectively. Table 14 provides the comprehensive results, which indicates that FADformer consistently outperforms state-of-the-art models.

**Robust of Our Debiasing Modules in Different Models.** To evaluate the robustness of our debiasing plugins, we conducted multiple runs of the model under different random seeds. As seen in Table 15, we select Timebridge and TQNet as baselines and incorporate the debiasing plugins

---

[2]https://www.kaggle.com/datasets/sai14karthik/nasdq-dataset.
[3]https://www.kaggle.com/datasets/sandeshbhat/wikipedia-webtraffic-201819.

Table 13: Extra results compared with more baselines. The input length is fixed to 96.

| Models | | FADformer (Ours) | | Amplifier (2025) | | Fredformer (2024) | | SAMformer (2024) | | CycleNet (2024a) | | SparseTSF (2024b) | | FilterNet (2024) | | FreDF (2025) | | FreTS (2023) | |
|---|---|---|---|---|---|---|---|---|---|---|---|---|---|---|---|---|---|---|---|
| Metric | | MSE | MAE | MSE | MAE | MSE | MAE | MSE | MAE | MSE | MAE | MSE | MAE | MSE | MAE | MSE | MAE | MSE | MAE |
| ETTh1 | 96 | 0.378 | 0.394 | 0.371 | 0.392 | 0.373 | 0.392 | 0.384 | 0.402 | 0.378 | 0.397 | 0.376 | 0.389 | 0.376 | 0.397 | 0.382 | 0.401 | 0.399 | 0.412 |
| | 192 | 0.431 | 0.422 | 0.425 | 0.422 | 0.433 | 0.420 | 0.439 | 0.430 | 0.441 | 0.431 | 0.426 | 0.419 | 0.445 | 0.430 | 0.430 | 0.427 | 0.453 | 0.443 |
| | 336 | 0.475 | 0.445 | 0.448 | 0.434 | 0.470 | 0.437 | 0.476 | 0.453 | 0.495 | 0.453 | 0.470 | 0.441 | 0.487 | 0.451 | 0.478 | 0.455 | 0.503 | 0.475 |
| | 720 | 0.489 | 0.475 | 0.476 | 0.464 | 0.467 | 0.456 | 0.531 | 0.502 | 0.502 | 0.473 | 0.482 | 0.469 | 0.491 | 0.469 | 0.463 | 0.462 | 0.595 | 0.565 |
| | Avg | 0.443 | 0.434 | 0.430 | 0.428 | 0.436 | 0.426 | 0.458 | 0.444 | 0.454 | 0.439 | 0.439 | 0.430 | 0.450 | 0.437 | 0.438 | 0.436 | 0.488 | 0.474 |
| ETTh2 | 96 | 0.293 | 0.340 | 0.283 | 0.337 | 0.293 | 0.342 | 0.298 | 0.358 | 0.299 | 0.344 | 0.308 | 0.354 | 0.292 | 0.343 | 0.288 | 0.336 | 0.350 | 0.403 |
| | 192 | 0.368 | 0.388 | 0.359 | 0.390 | 0.371 | 0.389 | 0.369 | 0.388 | 0.374 | 0.400 | 0.388 | 0.398 | 0.370 | 0.396 | 0.363 | 0.384 | 0.472 | 0.475 |
| | 336 | 0.419 | 0.427 | 0.390 | 0.415 | 0.382 | 0.409 | 0.381 | 0.406 | 0.425 | 0.436 | 0.431 | 0.442 | 0.420 | 0.432 | 0.428 | 0.431 | 0.564 | 0.528 |
| | 720 | 0.422 | 0.440 | 0.446 | 0.456 | 0.415 | 0.434 | 0.425 | 0.448 | 0.442 | 0.454 | 0.432 | 0.447 | 0.430 | 0.446 | 0.416 | 0.437 | 0.815 | 0.654 |
| | Avg | 0.376 | 0.399 | 0.370 | 0.400 | 0.365 | 0.394 | 0.368 | 0.400 | 0.385 | 0.409 | 0.390 | 0.410 | 0.378 | 0.404 | 0.374 | 0.397 | 0.550 | 0.515 |
| ETTm1 | 96 | 0.319 | 0.349 | 0.316 | 0.355 | 0.331 | 0.368 | 0.333 | 0.368 | 0.321 | 0.362 | 0.349 | 0.376 | 0.318 | 0.358 | 0.324 | 0.361 | 0.339 | 0.374 |
| | 192 | 0.370 | 0.379 | 0.361 | 0.381 | 0.365 | 0.389 | 0.367 | 0.385 | 0.361 | 0.381 | 0.380 | 0.393 | 0.364 | 0.383 | 0.373 | 0.386 | 0.382 | 0.397 |
| | 336 | 0.403 | 0.396 | 0.393 | 0.404 | 0.405 | 0.413 | 0.413 | 0.402 | 0.392 | 0.404 | 0.410 | 0.413 | 0.396 | 0.408 | 0.402 | 0.404 | 0.421 | 0.426 |
| | 720 | 0.469 | 0.434 | 0.456 | 0.440 | 0.463 | 0.448 | 0.478 | 0.449 | 0.448 | 0.440 | 0.473 | 0.447 | 0.456 | 0.443 | 0.468 | 0.444 | 0.485 | 0.462 |
| | Avg | 0.390 | 0.390 | 0.382 | 0.395 | 0.391 | 0.405 | 0.398 | 0.401 | 0.381 | 0.397 | 0.403 | 0.407 | 0.384 | 0.398 | 0.392 | 0.399 | 0.407 | 0.415 |
| ETTm2 | 96 | 0.174 | 0.252 | 0.178 | 0.261 | 0.177 | 0.259 | 0.178 | 0.278 | 0.164 | 0.246 | 0.180 | 0.262 | 0.175 | 0.257 | 0.173 | 0.252 | 0.190 | 0.282 |
| | 192 | 0.239 | 0.295 | 0.244 | 0.304 | 0.243 | 0.301 | 0.255 | 0.308 | 0.231 | 0.291 | 0.243 | 0.302 | 0.240 | 0.300 | 0.241 | 0.298 | 0.260 | 0.329 |
| | 336 | 0.303 | 0.335 | 0.309 | 0.346 | 0.302 | 0.340 | 0.308 | 0.349 | 0.284 | 0.328 | 0.301 | 0.338 | 0.298 | 0.339 | 0.301 | 0.337 | 0.373 | 0.405 |
| | 720 | 0.399 | 0.394 | 0.390 | 0.394 | 0.397 | 0.396 | 0.400 | 0.399 | 0.384 | 0.389 | 0.395 | 0.396 | 0.392 | 0.393 | 0.403 | 0.396 | 0.517 | 0.499 |
| | Avg | 0.279 | 0.319 | 0.280 | 0.326 | 0.280 | 0.324 | 0.285 | 0.334 | 0.266 | 0.314 | 0.280 | 0.325 | 0.276 | 0.322 | 0.280 | 0.321 | 0.335 | 0.379 |
| ECL | 96 | 0.141 | 0.229 | 0.147 | 0.242 | 0.147 | 0.241 | 0.155 | 0.255 | 0.136 | 0.230 | 0.197 | 0.269 | 0.147 | 0.242 | 0.144 | 0.233 | 0.189 | 0.277 |
| | 192 | 0.156 | 0.246 | 0.158 | 0.251 | 0.165 | 0.258 | 0.168 | 0.269 | 0.153 | 0.245 | 0.195 | 0.272 | 0.162 | 0.254 | 0.159 | 0.247 | 0.193 | 0.282 |
| | 336 | 0.171 | 0.260 | 0.175 | 0.271 | 0.177 | 0.273 | 0.184 | 0.282 | 0.170 | 0.264 | 0.209 | 0.287 | 0.177 | 0.271 | 0.172 | 0.263 | 0.207 | 0.296 |
| | 720 | 0.196 | 0.283 | 0.206 | 0.298 | 0.213 | 0.304 | 0.219 | 0.306 | 0.212 | 0.300 | 0.251 | 0.321 | 0.229 | 0.319 | 0.206 | 0.296 | 0.245 | 0.332 |
| | Avg | 0.166 | 0.255 | 0.172 | 0.266 | 0.176 | 0.269 | 0.182 | 0.278 | 0.168 | 0.260 | 0.213 | 0.287 | 0.179 | 0.272 | 0.170 | 0.260 | 0.209 | 0.297 |
| Traffic | 96 | 0.386 | 0.254 | 0.456 | 0.299 | 0.406 | 0.277 | 0.409 | 0.281 | 0.459 | 0.297 | 0.593 | 0.343 | 0.430 | 0.295 | 0.391 | 0.265 | 0.528 | 0.341 |
| | 192 | 0.403 | 0.261 | 0.472 | 0.318 | 0.426 | 0.290 | 0.420 | 0.289 | 0.457 | 0.295 | 0.561 | 0.343 | 0.447 | 0.298 | 0.411 | 0.273 | 0.531 | 0.338 |
| | 336 | 0.414 | 0.267 | 0.487 | 0.320 | 0.437 | 0.292 | 0.428 | 0.294 | 0.470 | 0.300 | 0.575 | 0.345 | 0.465 | 0.303 | 0.424 | 0.280 | 0.551 | 0.345 |
| | 720 | 0.445 | 0.283 | 0.517 | 0.332 | 0.462 | 0.305 | 0.458 | 0.302 | 0.502 | 0.314 | 0.622 | 0.347 | 0.497 | 0.320 | 0.461 | 0.298 | 0.598 | 0.367 |
| | Avg | 0.412 | 0.266 | 0.483 | 0.317 | 0.433 | 0.291 | 0.429 | 0.292 | 0.472 | 0.302 | 0.588 | 0.345 | 0.460 | 0.304 | 0.422 | 0.279 | 0.552 | 0.348 |
| Solar | 96 | 0.199 | 0.208 | 0.234 | 0.283 | 0.185 | 0.233 | 0.219 | 0.248 | 0.187 | 0.245 | 0.254 | 0.317 | 0.205 | 0.242 | 0.327 | 0.354 | 0.192 | 0.225 |
| | 192 | 0.223 | 0.228 | 0.237 | 0.259 | 0.227 | 0.253 | 0.237 | 0.269 | 0.205 | 0.261 | 0.255 | 0.306 | 0.233 | 0.265 | 0.350 | 0.353 | 0.229 | 0.252 |
| | 336 | 0.245 | 0.243 | 0.247 | 0.269 | 0.246 | 0.284 | 0.253 | 0.278 | 0.220 | 0.271 | 0.265 | 0.309 | 0.249 | 0.278 | 0.370 | 0.354 | 0.242 | 0.269 |
| | 720 | 0.247 | 0.242 | 0.246 | 0.270 | 0.247 | 0.276 | 0.262 | 0.284 | 0.220 | 0.263 | 0.262 | 0.303 | 0.253 | 0.281 | 0.289 | 0.293 | 0.240 | 0.272 |
| | Avg | 0.229 | 0.230 | 0.241 | 0.270 | 0.226 | 0.262 | 0.243 | 0.270 | 0.208 | 0.260 | 0.259 | 0.309 | 0.235 | 0.267 | 0.334 | 0.339 | 0.226 | 0.255 |
| Weather | 96 | 0.162 | 0.196 | 0.167 | 0.212 | 0.163 | 0.207 | 0.197 | 0.250 | 0.158 | 0.203 | 0.181 | 0.230 | 0.162 | 0.210 | 0.165 | 0.201 | 0.184 | 0.239 |
| | 192 | 0.212 | 0.244 | 0.215 | 0.251 | 0.211 | 0.251 | 0.236 | 0.277 | 0.207 | 0.248 | 0.225 | 0.266 | 0.213 | 0.256 | 0.218 | 0.252 | 0.223 | 0.275 |
| | 336 | 0.272 | 0.287 | 0.276 | 0.292 | 0.267 | 0.292 | 0.279 | 0.306 | 0.263 | 0.290 | 0.280 | 0.311 | 0.277 | 0.302 | 0.275 | 0.294 | 0.272 | 0.316 |
| | 720 | 0.351 | 0.342 | 0.352 | 0.346 | 0.343 | 0.341 | 0.346 | 0.344 | 0.344 | 0.344 | 0.359 | 0.364 | 0.352 | 0.349 | 0.373 | 0.363 | 0.340 | 0.363 |
| | Avg | 0.249 | 0.267 | 0.253 | 0.275 | 0.246 | 0.273 | 0.265 | 0.294 | 0.243 | 0.271 | 0.261 | 0.293 | 0.251 | 0.279 | 0.258 | 0.278 | 0.255 | 0.298 |
| Exchange | 96 | 0.084 | 0.203 | 0.087 | 0.206 | 0.088 | 0.212 | 0.161 | 0.308 | 0.090 | 0.210 | 0.191 | 0.314 | 0.091 | 0.211 | 0.092 | 0.213 | 0.086 | 0.212 |
| | 192 | 0.176 | 0.298 | 0.179 | 0.301 | 0.184 | 0.302 | 0.246 | 0.372 | 0.179 | 0.302 | 0.345 | 0.428 | 0.186 | 0.305 | 0.190 | 0.310 | 0.217 | 0.344 |
| | 336 | 0.329 | 0.415 | 0.361 | 0.437 | 0.342 | 0.429 | 0.368 | 0.453 | 0.365 | 0.438 | 0.981 | 0.754 | 0.380 | 0.449 | 0.368 | 0.441 | 0.415 | 0.475 |
| | 720 | 0.827 | 0.688 | 0.879 | 0.704 | 0.849 | 0.689 | 1.005 | 0.750 | 0.931 | 0.724 | 0.783 | 0.626 | 0.898 | 0.712 | 0.856 | 0.700 | 0.947 | 0.725 |
| | Avg | 0.354 | 0.401 | 0.377 | 0.412 | 0.366 | 0.408 | 0.445 | 0.471 | 0.391 | 0.419 | 0.575 | 0.531 | 0.389 | 0.419 | 0.377 | 0.416 | 0.416 | 0.439 |
| PEMS03 | 12 | 0.067 | 0.168 | 0.070 | 0.172 | 0.068 | 0.174 | 0.117 | 0.226 | 0.066 | 0.172 | 0.120 | 0.237 | 0.077 | 0.187 | 0.069 | 0.173 | 0.083 | 0.194 |
| | 24 | 0.087 | 0.192 | 0.090 | 0.200 | 0.093 | 0.202 | 0.235 | 0.324 | 0.089 | 0.201 | 0.185 | 0.296 | 0.112 | 0.224 | 0.098 | 0.207 | 0.127 | 0.241 |
| | 48 | 0.122 | 0.231 | 0.147 | 0.260 | 0.146 | 0.258 | 0.333 | 0.425 | 0.136 | 0.247 | 0.341 | 0.403 | 0.169 | 0.277 | 0.130 | 0.245 | 0.202 | 0.310 |
| | 96 | 0.155 | 0.264 | 0.217 | 0.323 | 0.228 | 0.330 | 0.457 | 0.515 | 0.182 | 0.282 | 0.591 | 0.537 | 0.220 | 0.322 | 0.165 | 0.272 | 0.265 | 0.365 |
| | Avg | 0.108 | 0.214 | 0.131 | 0.239 | 0.135 | 0.243 | 0.286 | 0.373 | 0.118 | 0.226 | 0.309 | 0.368 | 0.145 | 0.253 | 0.116 | 0.224 | 0.169 | 0.278 |
| PEMS04 | 12 | 0.075 | 0.180 | 0.082 | 0.190 | 0.085 | 0.189 | 0.129 | 0.239 | 0.078 | 0.186 | 0.139 | 0.258 | 0.082 | 0.190 | 0.092 | 0.204 | 0.097 | 0.209 |
| | 24 | 0.091 | 0.200 | 0.102 | 0.215 | 0.117 | 0.224 | 0.153 | 0.275 | 0.099 | 0.212 | 0.210 | 0.320 | 0.110 | 0.224 | 0.128 | 0.243 | 0.144 | 0.258 |
| | 48 | 0.118 | 0.227 | 0.151 | 0.269 | 0.174 | 0.276 | 0.229 | 0.326 | 0.133 | 0.248 | 0.375 | 0.431 | 0.160 | 0.276 | 0.213 | 0.295 | 0.223 | 0.328 |
| | 96 | 0.149 | 0.258 | 0.205 | 0.320 | 0.273 | 0.354 | 0.291 | 0.379 | 0.168 | 0.281 | 0.648 | 0.576 | 0.234 | 0.276 | 0.248 | 0.334 | 0.288 | 0.379 |
| | Avg | 0.108 | 0.216 | 0.135 | 0.249 | 0.162 | 0.261 | 0.201 | 0.305 | 0.120 | 0.232 | 0.343 | 0.396 | 0.146 | 0.258 | 0.170 | 0.269 | 0.188 | 0.294 |
| PEMS07 | 12 | 0.070 | 0.161 | 0.079 | 0.179 | 0.073 | 0.184 | 0.109 | 0.222 | 0.062 | 0.162 | 0.111 | 0.229 | 0.064 | 0.163 | 0.073 | 0.184 | 0.078 | 0.185 |
| | 24 | 0.088 | 0.182 | 0.094 | 0.196 | 0.092 | 0.192 | 0.150 | 0.262 | 0.086 | 0.192 | 0.176 | 0.288 | 0.093 | 0.200 | 0.111 | 0.219 | 0.127 | 0.239 |
| | 48 | 0.111 | 0.205 | 0.129 | 0.237 | 0.136 | 0.241 | 0.210 | 0.329 | 0.129 | 0.235 | 0.325 | 0.393 | 0.137 | 0.248 | 0.157 | 0.246 | 0.220 | 0.317 |
| | 96 | 0.138 | 0.232 | 0.185 | 0.291 | 0.197 | 0.298 | 0.246 | 0.367 | 0.176 | 0.268 | 0.533 | 0.513 | 0.198 | 0.306 | 0.239 | 0.278 | 0.316 | 0.386 |
| | Avg | 0.102 | 0.195 | 0.122 | 0.226 | 0.121 | 0.222 | 0.179 | 0.295 | 0.113 | 0.214 | 0.286 | 0.358 | 0.123 | 0.229 | 0.145 | 0.232 | 0.185 | 0.282 |
| PEMS08 | 12 | 0.077 | 0.176 | 0.079 | 0.182 | 0.081 | 0.185 | 0.122 | 0.233 | 0.082 | 0.186 | 0.132 | 0.238 | 0.080 | 0.182 | 0.083 | 0.185 | 0.096 | 0.204 |
| | 24 | 0.110 | 0.209 | 0.115 | 0.218 | 0.112 | 0.214 | 0.165 | 0.265 | 0.118 | 0.227 | 0.199 | 0.295 | 0.114 | 0.219 | 0.123 | 0.220 | 0.152 | 0.256 |
| | 48 | 0.174 | 0.257 | 0.192 | 0.294 | 0.174 | 0.267 | 0.215 | 0.293 | 0.170 | 0.268 | 0.356 | 0.393 | 0.184 | 0.284 | 0.167 | 0.259 | 0.247 | 0.331 |
| | 96 | 0.226 | 0.262 | 0.346 | 0.390 | 0.277 | 0.335 | 0.313 | 0.345 | 0.235 | 0.305 | 0.632 | 0.518 | 0.309 | 0.356 | 0.226 | 0.298 | 0.354 | 0.395 |
| | Avg | 0.147 | 0.226 | 0.183 | 0.271 | 0.161 | 0.250 | 0.204 | 0.284 | 0.151 | 0.247 | 0.330 | 0.361 | 0.172 | 0.260 | 0.150 | 0.241 | 0.212 | 0.297 |
| 1st Count | | 28 | 48 | 8 | 1 | 3 | 4 | 2 | 1 | 23 | 6 | 0 | 2 | 0 | 0 | 3 | 2 | 1 | 0 |

Table 14: Full results of multivariate time series forecasting across more datasets.

| Models | | FADformer (Ours) | | Amplifier (2025) | | Fredformer (2024) | | iTransformer (2024) | | SOFTS (2024) | | CycleNet (2024) | | FilterNet (2024) | | PatchTST (2023) | | FreTS (2023) | |
|---|---|---|---|---|---|---|---|---|---|---|---|---|---|---|---|---|---|---|---|
| Metrics | | MSE | MAE | MSE | MAE | MSE | MAE | MSE | MAE | MSE | MAE | MSE | MAE | MSE | MAE | MSE | MAE | MSE | MAE |
| NASDAQ | 24 | 0.120 | 0.218 | 0.153 | 0.275 | 0.137 | 0.237 | 0.127 | 0.234 | 0.119 | 0.216 | 0.135 | 0.247 | 0.174 | 0.294 | 0.127 | 0.224 | 0.198 | 0.299 |
| | 36 | 0.169 | 0.265 | 0.197 | 0.308 | 0.185 | 0.284 | 0.179 | 0.272 | 0.174 | 0.271 | 0.169 | 0.284 | 0.200 | 0.309 | 0.174 | 0.269 | 0.229 | 0.327 |
| | 48 | 0.208 | 0.299 | 0.239 | 0.342 | 0.231 | 0.319 | 0.225 | 0.314 | 0.238 | 0.325 | 0.196 | 0.309 | 0.302 | 0.391 | 0.225 | 0.314 | 0.268 | 0.353 |
| | 60 | 0.250 | 0.332 | 0.316 | 0.397 | 0.279 | 0.352 | 0.265 | 0.339 | 0.267 | 0.352 | 0.272 | 0.368 | 0.382 | 0.447 | 0.268 | 0.340 | 0.317 | 0.384 |
| | avg | 0.187 | 0.279 | 0.226 | 0.331 | 0.208 | 0.298 | 0.199 | 0.290 | 0.200 | 0.291 | 0.193 | 0.302 | 0.265 | 0.360 | 0.199 | 0.287 | 0.253 | 0.341 |
| Wiki | 24 | 6.809 | 0.435 | 8.023 | 0.612 | 6.844 | 0.468 | 6.834 | 0.491 | 6.624 | 0.484 | 6.819 | 0.484 | 6.882 | 0.502 | 6.858 | 0.430 | 8.023 | 0.612 |
| | 36 | 6.340 | 0.452 | 7.229 | 0.595 | 6.498 | 0.476 | 6.484 | 0.474 | 5.938 | 0.497 | 6.351 | 0.499 | 6.393 | 0.538 | 6.402 | 0.447 | 7.229 | 0.595 |
| | 48 | 5.899 | 0.464 | 7.185 | 0.640 | 6.530 | 0.480 | 5.918 | 0.487 | 5.874 | 0.519 | 5.978 | 0.521 | 5.946 | 0.547 | 5.956 | 0.452 | 7.184 | 0.641 |
| | 60 | 5.543 | 0.476 | 6.807 | 0.621 | 6.574 | 0.499 | 5.648 | 0.486 | 5.693 | 0.520 | 5.784 | 0.523 | 5.805 | 0.553 | 5.631 | 0.456 | 6.805 | 0.648 |
| | avg | 6.148 | 0.457 | 7.311 | 0.617 | 6.612 | 0.481 | 6.221 | 0.485 | 6.032 | 0.505 | 6.233 | 0.507 | 6.257 | 0.535 | 6.212 | 0.446 | 7.310 | 0.624 |
| Covid-19 | 24 | 4.455 | 1.233 | 5.133 | 1.394 | 4.715 | 1.231 | 4.799 | 1.531 | 6.587 | 1.528 | 4.860 | 1.394 | 5.133 | 1.342 | 5.528 | 1.450 | 5.634 | 1.442 |
| | 36 | 6.842 | 1.623 | 7.378 | 1.729 | 7.298 | 1.634 | 7.536 | 1.834 | 9.114 | 1.848 | 7.379 | 1.708 | 7.484 | 1.725 | 8.351 | 1.830 | 9.114 | 1.849 |
| | 48 | 10.210 | 2.010 | 11.013 | 2.227 | 10.141 | 2.067 | 10.051 | 2.133 | 12.804 | 2.135 | 11.013 | 2.106 | 11.333 | 2.119 | 11.259 | 2.114 | 10.941 | 2.033 |
| | 60 | 12.042 | 2.119 | 12.528 | 2.235 | 11.972 | 2.261 | 12.235 | 2.159 | 14.244 | 2.275 | 12.528 | 2.227 | 12.675 | 2.431 | 12.665 | 2.224 | 12.892 | 2.184 |
| | avg | 8.387 | 1.746 | 9.013 | 1.896 | 8.532 | 1.798 | 8.655 | 1.914 | 10.687 | 1.947 | 8.945 | 1.859 | 9.156 | 1.904 | 9.451 | 1.905 | 9.645 | 1.877 |
| Wind | 96 | 0.620 | 0.855 | 0.663 | 0.898 | 0.632 | 0.881 | 0.627 | 0.875 | 0.658 | 0.998 | 0.649 | 0.933 | 0.674 | 0.957 | 0.652 | 0.889 | 0.697 | 1.030 |
| | 192 | 0.711 | 1.032 | 0.774 | 1.267 | 0.715 | 1.034 | 0.719 | 1.066 | 0.752 | 1.214 | 0.743 | 1.097 | 0.768 | 1.147 | 0.746 | 1.076 | 0.809 | 1.276 |
| | 336 | 0.767 | 1.150 | 0.836 | 1.312 | 0.779 | 1.159 | 0.782 | 1.164 | 0.827 | 1.313 | 0.809 | 1.242 | 0.823 | 1.310 | 0.813 | 1.211 | 0.864 | 1.312 |
| | 720 | 0.807 | 1.232 | 0.865 | 1.428 | 0.817 | 1.234 | 0.821 | 1.245 | 0.887 | 1.474 | 0.835 | 1.281 | 0.865 | 1.333 | 0.851 | 1.308 | 0.893 | 1.417 |
| | Avg | 0.726 | 1.067 | 0.785 | 1.226 | 0.736 | 1.077 | 0.737 | 1.088 | 0.781 | 1.250 | 0.759 | 1.138 | 0.783 | 1.187 | 0.766 | 1.121 | 0.816 | 1.259 |
| CzeLan | 96 | 0.199 | 0.248 | 0.223 | 0.275 | 0.218 | 0.264 | 0.210 | 0.289 | 0.186 | 0.256 | 0.176 | 0.237 | 0.211 | 0.289 | 0.202 | 0.263 | 0.223 | 0.294 |
| | 192 | 0.214 | 0.258 | 0.245 | 0.307 | 0.270 | 0.301 | 0.236 | 0.304 | 0.226 | 0.290 | 0.215 | 0.269 | 0.252 | 0.322 | 0.220 | 0.288 | 0.243 | 0.312 |
| | 336 | 0.229 | 0.281 | 0.285 | 0.331 | 0.282 | 0.317 | 0.243 | 0.302 | 0.238 | 0.304 | 0.224 | 0.288 | 0.317 | 0.366 | 0.237 | 0.266 | 0.254 | 0.321 |
| | 720 | 0.246 | 0.298 | 0.304 | 0.346 | 0.309 | 0.327 | 0.273 | 0.325 | 0.265 | 0.323 | 0.262 | 0.337 | 0.358 | 0.392 | 0.254 | 0.302 | 0.273 | 0.335 |
| | Avg | 0.222 | 0.271 | 0.264 | 0.315 | 0.270 | 0.302 | 0.241 | 0.305 | 0.229 | 0.293 | 0.219 | 0.283 | 0.285 | 0.342 | 0.228 | 0.280 | 0.248 | 0.316 |
| ZafNoo | 96 | 0.436 | 0.411 | 0.476 | 0.418 | 0.484 | 0.404 | 0.453 | 0.428 | 0.432 | 0.419 | 0.466 | 0.412 | 0.445 | 0.426 | 0.434 | 0.424 | 0.504 | 0.468 |
| | 192 | 0.476 | 0.439 | 0.521 | 0.452 | 0.527 | 0.448 | 0.488 | 0.446 | 0.479 | 0.449 | 0.505 | 0.439 | 0.499 | 0.456 | 0.484 | 0.446 | 0.587 | 0.526 |
| | 336 | 0.512 | 0.455 | 0.563 | 0.488 | 0.559 | 0.454 | 0.524 | 0.479 | 0.521 | 0.470 | 0.541 | 0.465 | 0.532 | 0.480 | 0.518 | 0.464 | 0.635 | 0.623 |
| | 720 | 0.542 | 0.484 | 0.591 | 0.509 | 0.565 | 0.460 | 0.558 | 0.506 | 0.543 | 0.483 | 0.567 | 0.494 | 0.573 | 0.500 | 0.548 | 0.486 | 0.708 | 0.661 |
| | Avg | 0.492 | 0.447 | 0.538 | 0.467 | 0.534 | 0.442 | 0.506 | 0.465 | 0.494 | 0.455 | 0.520 | 0.453 | 0.512 | 0.466 | 0.496 | 0.455 | 0.609 | 0.570 |

into them without altering the original backbone or the training hyperparameters. The results indicate that our debiasing strategies consistently maintain low standard deviations across all settings, demonstrating their stability and reliability. Compared with the results in Table 9, the forecasting gains are subtle, which can be reasonably explained as follows: TimeBridge conditionally introduces non-stationarity into the attention layers, which alleviates the low-pass filtering effect to some extent and thus suppresses the feature over-smoothing issue. TQNet adopts a highly efficient single-layer attention layer architecture, which is minimally affected by the low-pass filtering of the attention mechanism. Leddam introduces a learnable decomposition and a dual attention module to separately capture dynamic trend information and seasonal dependencies, which cannot obtain complete high-frequency information to adjust attention and feature weights.

Table 15: Performance of our debiasing plugins on TimeBrideg and TQNet with different random seeds. The 'Mean' and 'Std' denote the average value and standard deviation, respectively.

| Models | | TimeBridge + Debiasing | | | | | | | | | | TQNet + Debiasing | | | | | | | | | |
|---|---|---|---|---|---|---|---|---|---|---|---|---|---|---|---|---|---|---|---|---|---|
| Setup | | Random Seed | | | | | | Mean | | Std | | Random Seed | | | | | | Mean | | Std | |
| | | 2023 | | 2024 | | 2025 | | | | | | 2024 | | 2025 | | 2026 | | | | | |
| Metric | | MSE | MAE | MSE | MAE | MSE | MAE | MSE | MAE | MSE | MAE | MSE | MAE | MSE | MAE | MSE | MAE | MSE | MAE | MSE | MAE |
| Weather | 96 | 0.142 | 0.183 | 0.141 | 0.183 | 0.142 | 0.183 | 0.142 | 0.183 | 0.0006 | 0.0006 | 0.155 | 0.199 | 0.154 | 0.198 | 0.155 | 0.199 | 0.155 | 0.199 | 0.0006 | 0.0006 |
| | 192 | 0.183 | 0.224 | 0.182 | 0.224 | 0.183 | 0.223 | 0.183 | 0.224 | 0.0006 | 0.0006 | 0.203 | 0.243 | 0.203 | 0.242 | 0.204 | 0.243 | 0.203 | 0.243 | 0.0006 | 0.0006 |
| | 336 | 0.234 | 0.265 | 0.232 | 0.264 | 0.233 | 0.264 | 0.233 | 0.264 | 0.0010 | 0.0006 | 0.259 | 0.285 | 0.259 | 0.284 | 0.260 | 0.285 | 0.259 | 0.285 | 0.0006 | 0.0006 |
| | 720 | 0.303 | 0.318 | 0.300 | 0.315 | 0.301 | 0.316 | 0.301 | 0.316 | 0.0015 | 0.0015 | 0.339 | 0.340 | 0.339 | 0.340 | 0.338 | 0.339 | 0.339 | 0.340 | 0.0006 | 0.0006 |
| Solar | 96 | 0.159 | 0.223 | 0.158 | 0.222 | 0.159 | 0.223 | 0.159 | 0.223 | 0.0006 | 0.0006 | 0.172 | 0.230 | 0.170 | 0.229 | 0.172 | 0.230 | 0.171 | 0.230 | 0.0012 | 0.0006 |
| | 192 | 0.174 | 0.235 | 0.173 | 0.234 | 0.174 | 0.234 | 0.174 | 0.234 | 0.0006 | 0.0006 | 0.197 | 0.256 | 0.196 | 0.256 | 0.197 | 0.255 | 0.197 | 0.256 | 0.0006 | 0.0006 |
| | 336 | 0.185 | 0.242 | 0.186 | 0.243 | 0.184 | 0.242 | 0.185 | 0.242 | 0.0010 | 0.0006 | 0.208 | 0.261 | 0.205 | 0.260 | 0.206 | 0.260 | 0.206 | 0.260 | 0.0015 | 0.0006 |
| | 720 | 0.196 | 0.250 | 0.194 | 0.248 | 0.196 | 0.249 | 0.195 | 0.249 | 0.0012 | 0.0010 | 0.206 | 0.269 | 0.205 | 0.268 | 0.207 | 0.270 | 0.206 | 0.269 | 0.0010 | 0.0010 |
| ECL | 96 | 0.118 | 0.213 | 0.118 | 0.212 | 0.118 | 0.213 | 0.118 | 0.213 | 0.0000 | 0.0006 | 0.132 | 0.227 | 0.132 | 0.227 | 0.133 | 0.227 | 0.132 | 0.227 | 0.0006 | 0.0000 |
| | 192 | 0.140 | 0.236 | 0.141 | 0.236 | 0.140 | 0.235 | 0.140 | 0.236 | 0.0006 | 0.0006 | 0.152 | 0.245 | 0.151 | 0.244 | 0.152 | 0.246 | 0.152 | 0.245 | 0.0006 | 0.0010 |
| | 336 | 0.155 | 0.250 | 0.156 | 0.252 | 0.154 | 0.249 | 0.155 | 0.250 | 0.0010 | 0.0015 | 0.165 | 0.263 | 0.165 | 0.264 | 0.164 | 0.263 | 0.165 | 0.263 | 0.0006 | 0.0010 |
| | 720 | 0.176 | 0.277 | 0.177 | 0.276 | 0.175 | 0.275 | 0.176 | 0.276 | 0.0015 | 0.0010 | 0.199 | 0.292 | 0.198 | 0.290 | 0.198 | 0.291 | 0.198 | 0.291 | 0.0006 | 0.0010 |
| Traffic | 96 | 0.338 | 0.239 | 0.337 | 0.238 | 0.338 | 0.239 | 0.338 | 0.239 | 0.0006 | 0.0006 | 0.403 | 0.257 | 0.403 | 0.256 | 0.404 | 0.257 | 0.403 | 0.257 | 0.0006 | 0.0006 |
| | 192 | 0.341 | 0.249 | 0.343 | 0.250 | 0.339 | 0.248 | 0.341 | 0.249 | 0.0020 | 0.0010 | 0.418 | 0.265 | 0.417 | 0.263 | 0.418 | 0.265 | 0.418 | 0.264 | 0.0006 | 0.0012 |
| | 336 | 0.360 | 0.255 | 0.361 | 0.256 | 0.358 | 0.254 | 0.360 | 0.255 | 0.0015 | 0.0010 | 0.439 | 0.274 | 0.440 | 0.274 | 0.438 | 0.275 | 0.439 | 0.274 | 0.0010 | 0.0006 |
| | 720 | 0.389 | 0.269 | 0.392 | 0.271 | 0.388 | 0.268 | 0.390 | 0.269 | 0.0021 | 0.0015 | 0.468 | 0.289 | 0.465 | 0.288 | 0.469 | 0.289 | 0.467 | 0.289 | 0.0021 | 0.0006 |

Table 16: Full results with lookback window length H ∈ {96, 336, 720}.

| Models | | FADformer | | SOFTS | | iTransformer | | FADformer | | SOFTS | | iTransformer | | FADformer | | SOFTS | | iTransformer | |
|---|---|---|---|---|---|---|---|---|---|---|---|---|---|---|---|---|---|---|---|
| Metric | | H=96 | | | | | | H=336 | | | | | | H=720 | | | | | |
| | | MSE | MAE | MSE | MAE | MSE | MAE | MSE | MAE | MSE | MAE | MSE | MAE | MSE | MAE | MSE | MAE | MSE | MAE |
| ETTm1 | 96 | 0.319 | 0.349 | 0.325 | 0.361 | 0.334 | 0.368 | 0.289 | 0.344 | 0.296 | 0.350 | 0.303 | 0.357 | 0.294 | 0.346 | 0.299 | 0.357 | 0.317 | 0.367 |
| | 192 | 0.370 | 0.379 | 0.370 | 0.385 | 0.377 | 0.391 | 0.331 | 0.369 | 0.360 | 0.394 | 0.345 | 0.383 | 0.326 | 0.366 | 0.342 | 0.381 | 0.347 | 0.385 |
| | 336 | 0.403 | 0.396 | 0.412 | 0.412 | 0.426 | 0.420 | 0.364 | 0.389 | 0.385 | 0.415 | 0.375 | 0.397 | 0.356 | 0.382 | 0.375 | 0.401 | 0.277 | 0.402 |
| | 720 | 0.469 | 0.434 | 0.472 | 0.448 | 0.491 | 0.459 | 0.425 | 0.423 | 0.449 | 0.463 | 0.435 | 0.432 | 0.410 | 0.411 | 0.441 | 0.443 | 0.429 | 0.431 |
| | Avg | **0.390** | **0.390** | 0.395 | 0.402 | 0.407 | 0.410 | **0.352** | **0.381** | 0.373 | 0.407 | 0.365 | 0.393 | **0.347** | **0.376** | 0.364 | 0.396 | 0.368 | 0.396 |
| ETTm2 | 96 | 0.174 | 0.252 | 0.180 | 0.261 | 0.180 | 0.264 | 0.165 | 0.247 | 0.174 | 0.259 | 0.184 | 0.273 | 0.163 | 0.249 | 0.181 | 0.272 | 0.187 | 0.278 |
| | 192 | 0.239 | 0.295 | 0.246 | 0.306 | 0.250 | 0.309 | 0.227 | 0.284 | 0.240 | 0.307 | 0.262 | 0.322 | 0.219 | 0.289 | 0.234 | 0.310 | 0.251 | 0.319 |
| | 336 | 0.303 | 0.335 | 0.316 | 0.350 | 0.311 | 0.348 | 0.287 | 0.320 | 0.295 | 0.342 | 0.307 | 0.351 | 0.268 | 0.326 | 0.284 | 0.342 | 0.307 | 0.355 |
| | 720 | 0.399 | 0.394 | 0.410 | 0.403 | 0.412 | 0.407 | 0.365 | 0.375 | 0.377 | 0.396 | 0.390 | 0.402 | 0.364 | 0.372 | 0.373 | 0.398 | 0.391 | 0.411 |
| | Avg | **0.279** | **0.319** | 0.288 | 0.330 | 0.288 | 0.332 | **0.261** | **0.307** | 0.272 | 0.326 | 0.286 | 0.337 | **0.254** | **0.309** | 0.268 | 0.331 | 0.284 | 0.341 |
| ECL | 96 | 0.141 | 0.229 | 0.143 | 0.233 | 0.148 | 0.240 | 0.134 | 0.221 | 0.135 | 0.230 | 0.138 | 0.237 | 0.128 | 0.220 | 0.137 | 0.232 | 0.142 | 0.243 |
| | 192 | 0.156 | 0.246 | 0.158 | 0.248 | 0.162 | 0.253 | 0.150 | 0.239 | 0.156 | 0.245 | 0.157 | 0.256 | 0.144 | 0.237 | 0.157 | 0.252 | 0.160 | 0.261 |
| | 336 | 0.171 | 0.260 | 0.178 | 0.269 | 0.178 | 0.269 | 0.165 | 0.254 | 0.170 | 0.263 | 0.172 | 0.267 | 0.160 | 0.252 | 0.172 | 0.268 | 0.179 | 0.281 |
| | 720 | 0.196 | 0.283 | 0.218 | 0.305 | 0.225 | 0.317 | 0.190 | 0.279 | 0.202 | 0.296 | 0.209 | 0.304 | 0.188 | 0.277 | 0.198 | 0.291 | 0.220 | 0.316 |
| | Avg | **0.166** | **0.255** | 0.174 | 0.264 | 0.178 | 0.270 | **0.160** | **0.248** | 0.166 | 0.259 | 0.169 | 0.266 | **0.155** | **0.247** | 0.166 | 0.261 | 0.175 | 0.275 |
| Traffic | 96 | 0.386 | 0.254 | 0.396 | 0.266 | 0.395 | 0.268 | 0.355 | 0.249 | 0.369 | 0.257 | 0.363 | 0.265 | 0.349 | 0.245 | 0.355 | 0.253 | 0.358 | 0.254 |
| | 192 | 0.403 | 0.261 | 0.413 | 0.274 | 0.417 | 0.276 | 0.376 | 0.258 | 0.383 | 0.272 | 0.385 | 0.273 | 0.367 | 0.251 | 0.369 | 0.261 | 0.375 | 0.263 |
| | 336 | 0.414 | 0.267 | 0.429 | 0.272 | 0.433 | 0.283 | 0.389 | 0.265 | 0.397 | 0.272 | 0.396 | 0.277 | 0.378 | 0.259 | 0.387 | 0.271 | 0.387 | 0.273 |
| | 720 | 0.445 | 0.283 | 0.447 | 0.287 | 0.467 | 0.302 | 0.424 | 0.278 | 0.427 | 0.316 | 0.435 | 0.312 | 0.419 | 0.268 | 0.429 | 0.286 | 0.418 | 0.292 |
| | Avg | **0.412** | **0.266** | 0.421 | 0.275 | 0.428 | 0.282 | **0.389** | **0.263** | 0.394 | 0.279 | 0.395 | 0.282 | **0.378** | **0.256** | 0.385 | 0.268 | 0.385 | 0.271 |
| Weather | 96 | 0.162 | 0.196 | 0.172 | 0.208 | 0.174 | 0.214 | 0.144 | 0.189 | 0.160 | 0.209 | 0.163 | 0.213 | 0.149 | 0.190 | 0.152 | 0.205 | 0.168 | 0.222 |
| | 192 | 0.212 | 0.244 | 0.221 | 0.256 | 0.221 | 0.254 | 0.189 | 0.235 | 0.204 | 0.250 | 0.203 | 0.250 | 0.192 | 0.238 | 0.199 | 0.251 | 0.209 | 0.256 |
| | 336 | 0.272 | 0.287 | 0.282 | 0.300 | 0.278 | 0.296 | 0.238 | 0.271 | 0.249 | 0.284 | 0.253 | 0.288 | 0.242 | 0.274 | 0.248 | 0.288 | 0.267 | 0.302 |
| | 720 | 0.351 | 0.342 | 0.358 | 0.350 | 0.358 | 0.347 | 0.315 | 0.339 | 0.324 | 0.335 | 0.326 | 0.338 | 0.312 | 0.333 | 0.322 | 0.343 | 0.337 | 0.352 |
| | Avg | **0.249** | **0.267** | 0.258 | 0.279 | 0.258 | 0.278 | **0.222** | **0.259** | 0.234 | 0.270 | 0.236 | 0.272 | **0.224** | **0.259** | 0.230 | 0.272 | 0.245 | 0.283 |

**Performance over Different Lookback Lengths.** To assess the performance of FADformer in capturing long-term temporal dependencies, we extend the lookback window sizes to 336 and 720 for further experimentation. As illustrated in Table 16, FADformer consistently delivers superior results across different forecasting horizons under these more demanding conditions. In contrast to models such as SOFTS and iTransformer, which exhibit significant performance degradation as the input length increases on the ECL and Weather datasets, FADformer maintains stable and accurate predictions, demonstrating robust temporal generalization.

