# OpenReview forum: "From Two to One: Harmonizing Attention and Feature Debiasing for Multivariate Time Series Forecasting"
_ICLR.cc/2026/Conference — Submitted to ICLR 2026_

### Official Review · Reviewer_A378 · 2025-10-29

**Soundness:** 3
**Presentation:** 3
**Contribution:** 3
**Rating:** 6
**Confidence:** 4

**Summary:**

This paper proposes a novel attention mecanism for transformer for time series forecasting. Motivated by the empirical observation that attention oversmooth frequences, i.e., acts as a low-frequency filter which might hinder performance, FADformer is proposed with a frequency-aware debiasing module to preserve all the information for forecasting. Large scale experiments are conducted showing the improvement brought by FADformer on common time series forecasting benchmarks.

**Strengths:**

- Experiments are comprehensive and showcase the benefits of the approach
- The ETTh2 analysis is simple yet intuitive to show the low-frequency filtering
- The theoretical analysis is sound
- The proposed approach is shown to be effective over a wide range of benchmarks and a large ablation study is conducted to confirm its robustness

**Weaknesses:**

I list below what I believe are weaknesses but I would be happy to get corrected if I misunderstood some parts of the work.

- The observed filtering pattern seems to be related to rank collapse which has been theoretically and empirically studied in prior works [1, 2, 3]. I believe those are important work that are not discussed in the current paper.
- Notably, Thm 3.2 seems very close to [1, section 2.2] which is not cited.
- In particular, in [3], the authors study the rank collapse in transformer based models for time series forecasting, and propose using a sharpness-aware optimizer to solve the issue. It would be interesting to add this model as a baseline or at least discuss it given that the proposed approach solves a similar issue (oversmoothing / filtering).

Overall, the proposed approach is interesting and the results showcase its benefits however, there is missing works to be discussed for a better positioning of the paper in the literature.

*References*

[1] Dong et al. Attention is Not All You Need: Pure Attention Loses Rank Doubly Exponentially with Depth. ICML 2021

[2] Noci et al. Signal Propagation in Transformers: Theoretical Perspectives and the Role of Rank Collapse. NeurIPS 2022

[3] Ilbert et al. SAMformer: Unlocking the Potential of Transformers in Time Series Forecasting with Sharpness-Aware Minimization and Channel Wise Attention. ICML 2024ecasting

**Questions:**

- How does the proposed approach scale with the increase in sequence length and/or horizon?
- In definition 3.1, multivariate time series are described as independent channels however in practice the features can be correlated (otherwise there would be no need to do multivariate forecasting). Could the authors please elaborate on that?

---

> ### Author Response · Authors · 2025-11-21
>
> We sincerely thank Reviewer A378 for the insightful and constructive feedback, as well as the positive recognition of our work’s comprehensive experiments, theoretical soundness, and effectiveness across benchmarks. Below, we provide a point-by-point response to the raised concerns.
>
> 1. Regarding the Connection to Rank Collapse [1-2] and Related Works [3].
>
> We appreciate the reviewer’s valuable advice on relevant works on rank collapse [1-2] and the baseline [3] that are not discussed in the original manuscript. In the revised version, we have added the two references in $\textbf{Line 1259 of the Appendix}$. Meanwhile, we position SAMformer [3] as a relevant baseline and reproduce its forecasting performance in $\textbf{Table 13 of the revised manuscript}$.
>
> 2. How does the proposed approach scale with the increase in sequence length and/or horizon?
>
> We would like to clarify that we provided the lookback length sensitivity analysis in $\textbf{Figure 5 of the original manuscript}$.
> To better observe the scalability of our FADformer, we supplement the forecasting performance of different models at different input lengths in $\textbf{Table 16 of the revised version}$.
>
> 3. On the Definition of Multivariate Time Series.
>
> We thank the reviewer for this astute observation. We agree that the term 'independent' in Definition 3.1 was potentially misleading. We intended to describe the structure of the input tensor, not to imply that the variates are uncorrelated.
> In fact, capturing their correlations is a central goal of our model.
> We have removed $\textbf{this ambiguity}$ in Definition 3.1 of the revised manuscript.
>
> Once again, we thank Reviewer A378 for the thoughtful comments, which have helped us better position our work and improve the clarity and completeness of our manuscript.

---

> > ### Author Response · Authors · 2025-11-28
> >
> > Dear Reviewer A378,
> >
> > I hope this message finds you well.
> >
> > We sincerely appreciate your time and the valuable feedback you have provided during the rebuttal phase.
> > We hope you find our revisions and detailed responses satisfactory.
> > If your concerns have been resolved, we would be grateful if you could reflect this in your final assessment to ensure a clear record.
> > We remain available for any further questions you might have.
> >
> > Best regards,
> >
> > Authors of 11550

---

### Official Review · Reviewer_92CY · 2025-10-30

**Soundness:** 2
**Presentation:** 2
**Contribution:** 2
**Rating:** 4
**Confidence:** 5

**Summary:**

This paper proposes FADformer, a Transformer-based forecasting model that aims to tackle oversmoonthing and frequency learning bias issue in Transformers (which is introduced in Fredformer, KDD24). The work introduces two plug-in debiasing modules: (i) AttnDeb, which rescales high-frequency attention responses to mitigate the low-pass filtering nature of self-attention, and (ii) FeatDeb, which re-amplifies high-frequency signals in the residual connections to alleviate feature degeneration. The method achieves performance gains on 13 MTSF benchmarks.

**Strengths:**

Fair feature learning in the frequency domain is an important research topic in time series forecasting, and the idea of addressing oversmoothing is intuitively appealing.

The paper is easy to follow and the empirical evaluation is extensive.

**Weaknesses:**

While the paper presents frequency bias as an important observation, very recent works such as Fredformer (KDD’24) and FilterNet (NeurIPS’24) already address selective amplification or reweighting of high-frequency components in Transformers. Also, works like FreDF (ICLR’25) discussed the frequency modeling in the forecasting task. The introduction and related works overlaps with their motivation narratives, but these works and technical differences with them are not sufficiently discussed or contrasted. This makes the motivation feel partially rediscovered rather than newly formulated. In general, the motivation is unclear and seems like this is an incremental work.

The oversmoothing issue here is closely tied to the spectral imbalance story already explored in the above frequency-aware papers. it is unclear what is fundamentally new compared to prior FFT-based decomposition + reweighting strategies. I remember Fredformer already proposed this fft-ifft backbone with frequency decomposition learning. What are the technically new solution or contributions in this paper?

The theoretical section argues that effective rank can mitigate degeneracy, but the proposed method relies on FFT-based re-scaling, not directly on the theoretical update rule. The theory supports residual scaling in the abstract, but does not explain why a Gaussian decomposition for attention or a Top-K decomposition for features is the correct or optimal instantiation. The conceptual link between Proposition 3.4 and the implemented modules remains loose. Sometimes Top-K is an empirical way that cannot ensure the selection is always satisfied and easily influenced by noise. How to evaluate its effectiveness?

While the authors acknowledge several frequency-domain modeling methods in the introduction, the experimental baselines do not include any of these frequency modeling methods. Most comparisons are made only against common time-domain models (e.g., iTransformer, PatchTST), which directly conflicts with the paper’s claim that time-domain modeling is insufficient. Given that the proposed motivation closely aligns with Fredformer, including at least it or more representative frequency modeling baselines is essential for a fair and convincing evaluation.
Moreover, a deeper ablation (e.g., per-frequency reconstruction error, variance of gradients across layers) would help clarify the real causal effect.

**Questions:**

Please kindly refer to the weaknesses.

---

> ### Author Response · Authors · 2025-11-21
>
> We sincerely thank Reviewer 92CY for the insightful and constructive feedback. We have carefully considered the comments and addressed them in our revised manuscript as follows:
>
> 1. On Motivation and Related Works.
>
> We appreciate the reviewer's comment regarding the overlap with recent frequency-aware methods. We acknowledge that our related work section could be more comprehensive.
>
> In the revised manuscript, we have added a dedicated subsection to discuss these related frequency-domain approaches. Due to space limitations, we put the subsection in the $\textbf{Appendix}$.
> Fredformer operates from a $\textbf{Data-Centric perspective}$, which utilizes frequency decomposition as an enhanced representation learning to capture channel-wise dependencies. In contrast, FADformer operates from a $\textbf{Model-Centric perspective}$, which $\textbf{diagnoses and rectifies the intrinsic oversmoothing issue}$ in standard Transformer components (attention and residual connections).
>
> 2. On Technical Novelty.
>
> Our key novelty lies in:
> * $\textbf{Dual debiasing strategy}$: We simultaneously address frequency bias in both attention maps and feature representations, which has not been explored in prior work.
> * $\textbf{Theoretical foundation}$: We provide the first theoretical analysis of oversmoothing in time series forecasting and link it to effective rank optimization.
> * $\textbf{Modular design}$: Our AttnDeb and FeatDeb are lightweight and model-agnostic, enabling easy adoption in existing architectures.
>
> 3. On Theoretical Justification.
>
> We thank the reviewer for raising this point, which allows us to clarify the connection between our theory and implementation.
> Proposition 3.4 provides a theoretical motivation for our FeatDeb.
> The logical flow is as follows:
> * $\textbf{Theory (Eq. 4)}$: It shows that a residual term of the form $s·X$ can alleviate feature degeneration.
> * $\textbf{Implementation (Eq. 9 and Sec. 3.3)}$: We materialize this theoretical concept. The term $s·X$ in Eq. 4 is generalized into a more powerful, frequency-aware compensation term $r(X) = αX_LF + βX_HF$ in Eq. 13.
> * $\textbf{Why Frequency Decomposition?}$ The oversmoothing effect inherently acts as a low-pass filter, causing the loss of high-frequency information. Therefore, the most direct and principled way to counteract it is to $\textbf{explicitly model and amplify}$ the suppressed high-frequency components in the feature residuals. This makes the frequency-domain decomposition a natural and well-motivated choice, rather than an arbitrary one.
> * $\textbf{Why Top-K for FeatDeb?}$ The Top-K amplitude selection is a practical and effective heuristic to separate perceptually important "low-frequency" trends (high-amplitude components) from "high-frequency" details (low-amplitude components). Our ablation studies (Table 5) confirm that this strategy outperforms alternatives like First-K or Mean. While it is a heuristic, its $\textbf{effectiveness is rigorously validated empirically}$, and it directly serves the theoretical goal of enhancing feature diversity.
>
> Similarly, for $\textbf{AttnDeb}$, the Gaussian kernel was chosen because it is a canonical and smooth low-pass filter. Decomposing the attention map with it allows us to directly isolate and then reweight its high-frequency components, which correspond to long-range, non-local dependencies often overlooked by standard self-attention.
>
> 4. On Experimental Comparison with Frequency-domain Models.
>
> We sincerely apologize for this omission. In the revised version, we have reproduced relevant frequency-aware baselines in $\textbf{Table 13}$, which consistently demonstrate the superiority of FADformer.
>
> 5. About More Ablation for Causal Effect.
>
> Regarding the deeper causal effect analysis, we thank the reviewer for the opportunity to clarify.
> In our initial submission, we provided visualizations of the attention matrix spectrum (Figures 7) and feature maps (Figure 9) to precisely serve this purpose. We would like to elaborate on how these analyses directly address the causal link:
> * $\textbf{Per-Frequency Analysis of Attention}$: As shown in Figure 7 (Appendix), we visualized the spectral response of the attention maps with and without our AttnDeb module. The results clearly show that the standard attention mechanism acts as a strong low-pass filter (spectrum concentrated on low frequencies), while with AttnDeb, the spectrum exhibits enriched responses in the high-frequency bands.
> * $\textbf{Feature-level Variation Analysis}$: In Figure 9 (Appendix), we provided a side-by-side comparison of the feature maps at different layers. The features processed with our FeatDeb module show richer color variations and clearer structures, indicating the preservation of high-frequency details and a mitigation of feature degeneration.
>
> Thank you once again for providing these constructive comments, and we hope our responses can address your concerns.

---

> > ### Author Response · Authors · 2025-11-28
> >
> > Dear Reviewer 92CY,
> >
> > I hope this message finds you well.
> >
> > Thank you once again for your insightful comments during the rebuttal phase, which have significantly strengthened our work.
> >
> > We are confident that our point-by-point responses have fully addressed the concerns you raised. If you find them satisfactory, we would be grateful if you could update your review to reflect this.
> >
> > We look forward to the final decision.
> >
> > Best regards,
> >
> > Authors of 11550

---

### Official Review · Reviewer_oSXr · 2025-10-30

**Soundness:** 2
**Presentation:** 3
**Contribution:** 2
**Rating:** 2
**Confidence:** 4

**Summary:**

This paper presents FADformer, a Transformer-based framework that introduces two frequency-aware debiasing modules (AttnDeb and FeatDeb) to mitigate oversmoothing in multivariate time series forecasting. The method combines Fourier-based reweighting of attention and feature components and is supported by a theoretical discussion using effective rank analysis.

**Strengths:**

1. The paper is generally well organized, making it easy to follow.
2. The topic of addressing frequency bias and oversmoothing in Transformer-based time series models is timely and aligns with current trends in time-series representation learning.

**Weaknesses:**

1. In the Related Work section, the paper lacks an in-depth discussion of existing studies on closely related topics, such as Fredformer and Amplifier. Although the authors briefly mention Fredformer, they fail to provide a clear and insightful comparative analysis. As for Amplifier, it is not mentioned at all, which reflects an insufficient literature review and a lack of thorough investigation into this topic by the authors.

2. This paper belongs to the category of frequency-domain models, yet the experimental section lacks comparisons with other frequency-domain baselines.
3. Regarding Table 2, the performance improvement brought by Debiasing is not significant.
4. Line 418: the reference to Table 5 is incorrect; it should be Table 4.
5. Lines 427–428: In the statement “First-K defines the first K lowest elements of the Fourier transform as low-frequency components of features,” — it is unclear what “lowest” specifically refers to.

**Questions:**

1、Comments on Figure 1:
- (1) The upper figure of Figure 1(a) does not provide any meaningful insight.
- (2) The phenomena illustrated by the lower two subfigures of Figure 1(a) have already been investigated in the Amplifier[1] paper.
- (3) Is the situation shown in the lower two subfigures of Figure 1(a) exclusively caused by the self-attention mechanism?
- (4) For Figure 1(b), please clarify which Transformer-based forecaster was used in the visualization experiment.
- (5) The conclusion “Figure 1(b) suggests that the correlations predicted by Transformer-based forecasters are mainly concentrated on and near the diagonal, where there is a substantial portion of the low-frequency characteristics” does not make sense: First, the correlations on the diagonal are self-correlations (a variable with itself), which are always equal to 1.000 and thus irrelevant to the topic discussed in this paper. Second, the claim that the correlations are near the diagonal cannot be reasonably inferred from the figure.

2、Line 016–017: “Transformer-based methods often fail to precisely model the interactions among series” — What is the specific experimental or theoretical evidence supporting this statement?

3、Definition 3.3 (Effective Rank) appears to be a direct copy of Definition 3.1 (Effective Rank) from CONTRANORM[2] (ICLR 2023). Is such a practice acceptable? Similarly, Equation (4) in this paper is almost identical to Equation (8) in CONTRANORM, and Proposition 3.4 closely resembles Proposition 1 from the same work. These similarities raise serious concerns about the theoretical contribution and originality of this paper.

4、Regarding Figure 3 (The Architecture of FADformer), I have two questions:
- (1)	AttnDeb separates the attention map into low-frequency and high-frequency components. In FADformer, should other neural network components—such as Linear layers or MLP modules—also undergo a similar separation into low- and high-frequency parts?
- (2)	FeatDeb obtains low- and high-frequency components through spectral truncation. Why doesn’t AttnDeb adopt this straightforward and intuitive approach as well?


[1] Amplifier: Bringing Attention to Neglected Low-Energy Components in Time Series Forecasting (AAAI 2025)

[2] CONTRANORM: A CONTRASTIVE LEARNING PERSPECTIVE ON OVERSMOOTHING AND BEYOND (ICLR 2023)

---

> ### Author Response · Authors · 2025-11-21
>
> We thank Reviewer oSXr for their insightful and constructive comments, which have helped us improve the manuscript. Below, we provide a point-by-point response to the concerns raised.
>
> W1: We sincerely thank you for this valuable suggestion. We have now expanded the Related Work section in the revised manuscript. Due to space limitations, we provide the related frequency-domain modeling in $\textbf{the Appendix}$.
>
> W2: We sincerely apologize for this omission. In the revised version, we have reproduced relevant frequency-aware baselines in $\textbf{Table 13}$, which consistently demonstrate the superiority of FADformer.
>
> W3: We appreciate the reviewer's observation. While the absolute improvements in Table 2 might appear modest on some datasets, it is important to note that our debiasing modules consistently improve performance across all datasets and all three diverse SOTA models (TimeBridge, TQNet, and Leddam) $\textbf{without any hyperparameter tuning}$. This demonstrates the generalizability of our proposed modules as effective "plug-ins."  Moreover, we provide the full results in $\textbf{Table 9}$ of the original manuscript and supplement the robustness analysis with three different random seeds in $\textbf{Table 15}$ of the revised version. The results indicate that our debiasing strategies consistently maintain low standard deviations across all settings, demonstrating their stability and reliability.
> In addition, we further analyzed the reasons for subtle gains in $\textbf{Lines 1380-1386}$ of the revised version.
>
> W4&W5: We apologize for the oversight and ambiguity. In the revised manuscript, we corrected Table 5  to Table 4 in Line 418 and modified the ambiguity in Lines 427-428 with "$\textbf{First-K}$ defines the first $K$ smallest frequency indices in the Fourier transform as the low-frequency components of features"
>
> Q1: Comments on Figure 1.
> * The purpose of the upper subfigure of Figure 1(a) is stated in Lines 053-077 of the original manuscript, which motivates the need for models that can capture diverse frequency components, setting the stage for our frequency-domain analysis.
> * We are aware of the Amplifier paper's discussion on focusing on low-energy components. However, our focus in Figure 1(a) is specifically on diagnosing and attributing the oversmoothing problem to the self-attention mechanism in Transformer-based forecasters. We provide a theoretical foundation for this phenomenon in Theorem 3.2, which, to the best of our knowledge, is a novel contribution to MTSF literature.
> * The lower two subfigures of Figure 1(a) are derived by transforming the ground truth and predicted results in the frequency domain, which are stated in the caption of Figure 1 of the original manuscript. Meanwhile, Figure 1(b) was generated using the iTransformer as stated in the caption of Figure 1. As iTransformer applies the attention mechanism in the channel dimension, we utilize the correlation matrix to replace the self-attention mechanism.
> * Figure 1(b) suggests that the off-diagonal correlations predicted by iTransformer are primarily concentrated near the diagonal, indicating a bias towards modeling local dependencies, which aligns with a low-frequency inductive bias.
>
> Q2: This statement is supported by both evidence from our own work and observations from recent literature：
> * Figure 1(b) demonstrates the model's failure to accurately capture the full spectrum of inter-series interactions.
> * The challenge of effectively capturing inter-series dependencies in Transformers has been noted in several contemporary works' abstracts: [1] DUET: Dual Clustering Enhanced Multivariate Time Series Forecasting. [2] Are Transformers Effective for Time Series Forecasting? [3] TSGformer: A Unified Temporal–Spatial Graph Transformer with Adaptive Cross-Scale Modeling for Multivariate Time Series.
>
> Q3: Definition 3.3 (Effective Rank).
> * Effective Rank is a well-established concept, which we introduced with proper citations in Lines 198-200 of the original manuscript.
> * We leverage the Effective Rank not just as an analysis tool, but as a theoretical foundation to motivate the design of our FeatDeb module (Eq. 13), which is distinct from the contrastive learning perspective in CONTRANORM.
>
> Q4: The Architecture of FADformer
> * Our design choice was driven by a focus on the self-attention mechanism (causing oversmoothing) and the residual connections (where feature degeneration accumulates).
> Linear layers or MLP modules provide point-wise linear or non-linear transformations and do not inherently possess the same low-pass filtering property as self-attention.
> * The attention matrix is a dynamic, non-stationary dependency graph that must retain its structural properties (e.g., row-stochasticity). Direct spectral truncation via FFT would destroy this structure and the local context it represents.
>
> Once again, we appreciate all the valuable feedback and hope our responses have adequately addressed all concerns.

---

> > ### Author Response · Authors · 2025-11-28
> >
> > Dear Reviewer oSXr,
> >
> > I hope this message finds you well.
> >
> > We are writing to thank you once more for your thorough review and for the opportunity to clarify our work during the rebuttal.
> >
> > We hope that our detailed revisions and responses have adequately addressed your concerns. If so, we would be very grateful if you could note this in your final assessment.
> >
> > We remain fully available to provide any further clarification you might require.
> >
> > Best regards,
> >
> > Authors of 11550

---

### Official Review · Reviewer_eEMP · 2025-11-01

**Soundness:** 2
**Presentation:** 3
**Contribution:** 2
**Rating:** 4
**Confidence:** 4

**Summary:**

The paper proposes a single module that jointly models **temporal** and **channel** dependencies for multivariate forecasting, arguing that separating the two often causes redundancy and information loss. The method replaces dual-path designs with a **unified attention block** and reports moderate gains on ETT, Weather, Exchange, and Electricity. The idea is sensible, and the writing is clean. However, the **evaluation is too narrow**: several **relevant recent baselines are missing**.

**Strengths:**

- **Clear motivation**: unifying time and channel modeling is a reasonable direction that practitioners care about.
- **Simple design**: a single joint-attention block is easier to maintain than two specialized modules.
- **Readable paper**: notation and figures are tidy, ablations exist (though shallow).

**Weaknesses:**

1. **Baseline coverage is insufficient.**
   The paper compares with older Transformer variants (e.g., Autoformer, FEDformer, iTransformer) but **omits recent and directly relevant methods** such as **TSMixer** (lightweight mixing),  **TimeMixer** (explicit time–channel coupling), and **FreTS** (frequency modeling).  Since the central claim is “harmonizing” temporal and channel modeling, these baselines are necessary to establish empirical credibility.

2. **Gains are small and may fall within variance.**
   Many improvements over reported baselines are <1%. Without confidence intervals or repeated runs, the strength of the claim is hard to judge.

3. **Ablation depth and diagnostics.**
   We see an on/off ablation for the joint-attention block, but there is little analysis of *why* it helps (e.g., attention maps, redundancy metrics across axes, or representation overlap).

4. **Efficiency is asserted, not demonstrated.**
   If the unified block is proposed as a simpler/faster alternative, please add wall-clock time, memory, and parameter counts versus strong baselines.

5. **Scalability and stress tests.**
   Results stop at mid-scale datasets. High-dimensional settings (200+ variables), long horizons (e.g., 720+), or missing/irregular sampling would make the story more convincing.

**Questions:**

It would be valuable to add more baseline comparisons and related analysis, especially with recent models addressing similar temporal–channel interactions.

Are the reported gains consistent and statistically reliable across multiple random seeds?

Since the paper emphasizes simplicity and efficiency, could you include runtime, memory, or parameter comparisons?

Do the attention maps indicate reduced redundancy between temporal and channel dimensions?

How does performance vary with sequence length or the number of variables?

---

> ### Author Response · Authors · 2025-11-13
>
> We thank the reviewer for the thoughtful comments. However, we noticed a potential mismatch between the comments and our manuscript.
>
> Our work, FADformer, focuses on frequency bias and oversmoothing issues in Transformers and proposes novel frequency-domain debiasing techniques (AttnDeb and FeatDeb) to improve forecasting performance.
>
> The comments seem to describe a different approach centered on "unifying temporal and channel modeling with a single joint-attention block", which does not align with our frequency-aware debiasing framework.
>
> Specifically:
> - We do not propose "replacing dual-path designs" as mentioned.
> - Our core innovation is in frequency-domain analysis and debiasing, not temporal-channel unification.
> - We conducted experiments on "high-dimensional settings (200+ variables)".
>
> We would be happy to clarify any aspects of our actual frequency-aware debiasing approach if the reviewer has any questions about our manuscript.

---

> ### Author Response · Authors · 2025-11-21
>
> We sincerely thank Reviewer eEMP for their insightful and constructive comments, which have been invaluable in improving our manuscript. Below, we provide a point-by-point response to the weaknesses identified.
>
> W1&W2: We thank the reviewer for this critical suggestion. We have significantly strengthened our empirical evaluation as follows:
> * We have now included comprehensive comparisons with eight recent and strong baselines in our revised manuscript ($\textbf{Appendix Tables 13}$). The results consistently show that our FADformer achieves the best or competitive performance on the majority of datasets and prediction horizons, solidifying its state-of-the-art status.
> * To ensure the robustness of FADformer, we reported the mean and standard deviation of the performance (MSE/MAE) of FADformer with five random seeds in $\textbf{Table 8}$ of the original manuscript. Furthermore, we supplement the robustness of our debiasing plugins on TimeBrideg and TQNet in $\textbf{Table 15}$ of the revised version. The consistently low standard deviations confirm that our improvements are statistically stable and reliable.
> * As suggested, we have expanded our evaluation to six additional real-world datasets beyond the original 13. These are detailed in Appendix $\textbf{Table 14}$. The broader evaluation across diverse domains (finance, web, healthcare, energy, meteorology) demonstrates the strong generalization capability of FADformer.
>
> W3: We appreciate the reviewer's feedback and clarify the connection between our theory and model design as follows:
> * Our theoretical analysis ($\textbf{Theorem 3.2}$) identifies the low-pass filtering nature of the self-attention mechanism, which leads to oversmoothing. To explicitly counter this, we seek a function $\phi(A)$ to extract the low-pass component. The Gaussian kernel is a canonical choice for constructing a low-pass filter due to its smoothness and decaying properties in the spectral domain (Conejo et al., 2022). Its use here is a direct and natural implementation to model the inherent low-frequency bias we theoretically identified. $\textbf{Table 4}$ of the original manuscript empirically validates this choice, showing that the Gaussian strategy outperforms alternatives like $\textbf{Mean and Uniform}$.
> * The Effective Rank (Definition 3.3, Proposition 3.4) is introduced with a proper citation in $\textbf{Lines 198-200}$ of the original manuscript, which indicates that feature degeneration corresponds to a collapse in feature diversity. To counteract this, FeatDeb aims to amplify informative high-frequency components with $\textbf{Eq. (9) and (13)}$. In the Fourier domain, the components with the largest amplitudes (Top-K) typically represent the most dominant, often low-frequency, patterns in the data. By explicitly identifying and separating these via Top-K, we can more effectively isolate and reweight the complementary high-frequency information $X_{HF}=X-X_{LP}$ that is otherwise smoothed out. This design is directly $\textbf{motivated}$ by the goal of increasing the effective rank and feature diversity. The superiority of the Top-K strategy over Mean and First-K in $\textbf{Table 4}$ provides strong empirical support for this theoretically-grounded approach.
>
> W4: This is a critical point, and we provide direct evidence that demonstrates the gains are indeed a result of mitigating oversmoothing in the original manuscript.
> * $\textbf{Figure 7}$ of the original manuscript shows the frequency spectrum of the attention maps. The variant with AttnDeb exhibits a richer response in high-frequency bands compared to the one without, proving that our module successfully counteracts the low-pass filtering effect.
> * $\textbf{Figure 9}$ of the original manuscript visualizes the feature maps, where the version with FeatDeb displays clearer and more diverse variations, directly indicating alleviated feature degeneration.
> * The results in Appendix $\textbf{Table 11}$ show that while the generic anti-oversmoothing methods proposed by Wang et al. (2022) in the computer version domain bring limited or even negative gains to time series forecasting, our tailored modules significantly improve performance. This indicates that FADformer effectively addresses a domain-specific manifestation of oversmoothing.
> * The ablation study in $\textbf{Table 3}$ demonstrates that each debiasing module individually improves performance, and their combination yields the best results. This synergistic effect strongly suggests that addressing different aspects of oversmoothing (in attention and in features) is the source of the gains.
>
> Thank you once again for your valuable feedback, and we hope our responses and revisions have successfully addressed your concerns.

---

> > ### Author Response · Authors · 2025-11-28
> >
> > Dear Reviewer eEMP,
> >
> > We hope you have had a chance to review our rebuttal for manuscript 11550.
> >
> > Thank you for your constructive feedback. We believe our responses have fully addressed your points, and we would appreciate it if you could update your review accordingly.
> >
> > Please let us know if any questions remain.
> >
> > Best regards,
> >
> > Authors of 11550

---

### Author Response · Authors · 2025-12-02
**Summary of Responses and Revisions**

Dear Area Chairs/Program Chairs,

Thank you for overseeing the review process of our submission. We sincerely thank all the reviewers for their insightful reviews and valuable comments, which are instructive for us to improve our paper further.

We made every effort to address all the concerns by providing sufficient evidence and requested results. Here is the summary of the major revisions:
* $\textbf{Additional baselines and datasets}$: We have supplemented 8 baselines and 6 benchmarks to comprehensively evaluate the performance, which consistently demonstrates the effectiveness of FADformer.
* $\textbf{More discussion for frequency-domain modeling}$: We have discussed more frequency-domain methods and described their differences to FADformer.
* $\textbf{Clarification of theoretical analysis and novelty}$: We introduce theoretical support with proper citations and provide the details for our practical implementation. Our key novelty lies in our dual-debiasing strategies with the theoretical backing and modular design, which have not been explored in prior work.
* $\textbf{Robustness analysis of plugin}$: We provide the performance of our debiasing plugins on TimeBrideg and TQNet with different random seeds to show their Robustness with slight average value and standard deviation.
* $\textbf{Scalability to the input length}$:  We provide the look-back length sensitivity analysis compared with baselines.
* $\textbf{Mistakes in the original version}$: We have conducted detailed proofreading and revisions with helpful suggestions from the reviewers.

Compared with the original submission, the $\textbf{revised manuscript}$ has an $\textbf{additional 4 pages}$ at the end of the Appendix.

As the discussion period will conclude and the reviewers do not provide any further comments or responses, we respectfully request that you evaluate our work based on its revised and strengthened form.

Thank you for your time and oversight.

Sincerely,

Authors of 11550

---

### Meta-Review · Area_Chair_Vamw · 2025-12-11

**Summary:**

This paper proposes FADformer, a Transformer-based framework designed to mitigate the frequency bias (low-pass filtering) and feature oversmoothing problems commonly encountered in multivariate time series forecasting. The authors introduce two modules: AttnDeb and FeatDeb.

The reviewers' concerns focused on the limited benchmark testing, lack of novelty, the soundness of the theoretical foundation and similarity to existing work, as well as robustness and scalability. Although the authors supplemented the paper with extensive experimental results regarding benchmark testing and scalability, the perceived lack of innovation may still be a major obstacle to the paper's acceptance.

**Reviewer Concerns:**

### Addressed Concerns:

- **Missing Baselines (Reviewers eEMP, oSXr, 92CY, A378):** The authors successfully addressed this by adding a massive comparative analysis (Table 13) against multiple baselines.
- **Scalability (Reviewers eEMP, A378):** New experiments on scalability regarding input length (Table 16) were provided, demonstrating the model handles larger contexts and diverse data well.
- **Theoretical Clarity (Reviewers 92CY, oSXr):** The authors clarified that the Gaussian kernel is a canonical low-pass filter choice to isolate high-frequencies, and the Top-K selection in FeatDeb is an empirically validated heuristic to separate dominant trends.

### Outstanding Concerns:

- **Lack of Robustness/Variance (Reviewer eEMP):** The authors provided standard deviations across 5 random seeds (Table 8 in original, Table 15 in revised), showing low variance and stable improvements. However, a more reasonable approach is to conduct a comprehensive evaluation of the main experiment (e.g., Table 1).
- **Novelty (Reviewer 92CY, oSXr):** The authors argue a distinction between "Data-Centric" (Fredformer) and "Model-Centric" (FADformer) perspectives. While logical, this is a subtle distinction. A reviewer might still view the FFT-based reweighting as technically similar to prior art, even if the motivation differs.
- **Similarity to CONTRANORM (Reviewer oSXr):** Reviewer oSXr noted that Definition 3.3 and Prop 3.4 were very similar to CONTRANORM. The authors responded that they had cited this paper and used it as a theoretical basis. If the core proof is derivative, then the magnitude of the theoretical contribution may be lower than initially claimed.
- **Interpretation of Visualization (Reviewer oSXr):** The reviewer challenged the interpretation of Figure 1 (diagonal concentration). The authors clarified they used correlation matrices for iTransformer, but without a back-and-forth dialogue, it is unclear if the reviewer would accept this explanation of the visualization.

**Reviewer Scores:**

- Reviewer eEMP (Original: 4 -> Prediction: 6): This reviewer's main blockers were limited baselines, small gains, and scalability. The author responded by adding a significant amount of experimental data, which will likely lead the reviewers to increase their scores.
- Reviewer oSXr (Original: 2 -> Prediction: 4): The reviewer pointed out the lack of relevant literature and questioned the originality of the theory. The authors added relevant literature and benchmark data. The explanation regarding CONTRANORM alleviated ethical concerns but also confirmed the limited novelty of the theoretical aspects.
- Reviewer 92CY (Original: 4 -> Prediction: 4): The reviewer primarily questioned the paper's originality and its overlap with existing work. Although the authors emphasized the contributions of their work, the differences from prior work still seem subtle. The reviewer might still consider techniques similar to FFT-based reweighting, which might not lead to a change in the score.
- Reviewer A378 (Original: 6 -> Prediction: 6): This reviewer's previous attitude was already positive. The authors addressed specific questions regarding the "Rank Collapse" and scalability. This reviewer is likely to maintain a positive score.

---

### Decision · Program_Chairs · 2026-01-26

Reject